# Enhancing Last-Mile Logistics: AI-Driven Fleet Optimization, Mixed Reality, and Large Language Model Assistants for Warehouse Operations

**DOI:** 10.3390/s25092696

**Published:** 2025-04-24

**Authors:** Saverio Ieva, Ivano Bilenchi, Filippo Gramegna, Agnese Pinto, Floriano Scioscia, Michele Ruta, Giuseppe Loseto

**Affiliations:** 1Department of Electrical and Information Engineering, Polytechnic University of Bari, Via E. Orabona 4, 70125 Bari, Italy; saverio.ieva@poliba.it (S.I.); ivano.bilenchi@poliba.it (I.B.); filippo.gramegna@poliba.it (F.G.); agnese.pinto@poliba.it (A.P.); michele.ruta@poliba.it (M.R.); 2Department of Engineering, LUM University “Giuseppe Degennaro”, Strada Statale 100 km 18, 70010 Casamassima, Italy; loseto@lum.it

**Keywords:** last-mile logistics, last-mile delivery, fleet optimization, vehicle routing, mixed reality, large language models, conversational assistants, knowledge graphs, process innovation, process optimization

## Abstract

Due to the rapid expansion of e-commerce and urbanization, Last-Mile Delivery (LMD) faces increasing challenges related to cost, timeliness, and sustainability. Artificial intelligence (AI) techniques are widely used to optimize fleet management, while augmented and mixed reality (AR/MR) technologies are being adopted to enhance warehouse operations. However, existing approaches often treat these aspects in isolation, missing opportunities for optimization and operational efficiency gains through improved information visibility across different roles in the logistics workforce. This work proposes the adoption of novel technological solutions integrated in an LMD framework that combines AI-based optimization of shipment allocation and vehicle route planning with a knowledge graph (KG)-driven decision support system. Additionally, the paper discusses the exploitation of relevant recent tools, including large language model (LLM)-powered conversational assistants for managers and operators and MR-based headset interfaces supporting warehouse operators by providing real-time data and enabling direct interaction with the system through virtual contextual UI elements. The framework prioritizes the customizability of AI algorithms and real-time information sharing between stakeholders. An experiment with a system prototype in the Apulia region is presented to evaluate the feasibility of the system in a realistic logistics scenario, highlighting its potential to enhance coordination and efficiency in LMD operations. The results suggest the usefulness of the approach while also identifying benefits and challenges in real-world applications.

## 1. Introduction

Last-Mile Delivery (LMD) is the final phase of outbound logistics, where goods are transported from warehouses or distribution hubs to private households. The global LMD market is expected to grow by over 85% between 2020 and 2027, increasing from 108 billion to more than 200 billion U.S. dollars [1]. The rapid expansion of e-commerce and urbanization has intensified the challenges associated with LMD, including rising operational costs, strict delivery time constraints, and environmental concerns [2]. In Europe, e-commerce has experienced consistent year-over-year growth since at least 2015, with a significant surge driven by the COVID-19 pandemic [3]. Simultaneously, urbanization has continued to expand, with over 70% of the population residing in cities as of 2000, increasing by approximately 1% every five years, with even steeper growth observed in Asia and Africa [4]. As a result, optimizing LMD and fleet operations has become a critical priority for modern logistics in smart city contexts, requiring innovative approaches to ensure cost-effective and timely deliveries [5].

To address these complexities, researchers and industry practitioners have increasingly turned to Artificial Intelligence (AI)-based techniques for optimizing shipment management in urban environments [6]. Several global companies have also underlined the impact of AI in logistics as evidenced in recent technology trend reports published by DHL Group [7] and FedEx [8]. Particularly, two of the primary challenges in LMD are warehouse management and fleet planning [9]. Warehouse operations encompass tasks such as product picking, packing, labeling, and vehicle loading, while fleet planning involves vehicle selection, load balancing, route optimization, and driver scheduling [10]. As confirmed in [6], however, existing AI-driven solutions often target these problems in isolation, leading to suboptimal coordination and missed opportunities for efficiency improvements.

A more integrated approach is needed to enhance decision-making across warehouse and fleet operations. Semantic web standards for knowledge representation serve as essential enablers in decision support for several real-world domains, including Industry 4.0 [11] and its human-centric evolution known as Industry 5.0 [12], driving assistance [13], and vehicular networks [14]. These experiences can be leveraged to integrate innovative technologies into LMD scenarios for structuring and interlinking critical logistics information. To this end, a Knowledge Graph (KG) can provide a unified and dynamically updated semantic layer for managing fleet configurations, shipment details, and warehouse layouts. As highlighted in [15], by embedding structured domain knowledge, the KG enhances interoperability between system components and supports automated reasoning to improve decision-making. Additionally, conversational assistants based on Large Language Models (LLMs) can facilitate real-time human–machine interaction [16], leveraging Retrieval Augmented Generation (RAG) techniques to provide context-aware responses and support operators [17] in executing warehouse and fleet management tasks.

Beyond AI-based decision-making and decision support, fostering seamless coordination between fleet managers and warehouse operators is essential. Web-based distributed architectures provide a foundational framework for enabling real-time data exchange, while advanced Mixed Reality (MR) interfaces can further enhance operational efficiency [18]. Head-Mounted Displays (HMDs) allow warehouse operators to visualize shipment allocations as digital overlays in the physical environment, reducing cognitive load and minimizing errors [19]. These systems enable operators to interact naturally with the logistics workflow, updating inventory and vehicle load status through intuitive hands-free interfaces. Moreover, MR techniques have shown promising synergy with semantic web technologies [20], which can be leveraged to complement visual overlays with semantically enriched information derived from sensors disseminated in the environment, allowing for automated reasoning over contextual data to support more informed and proactive decision-making [21]. Nevertheless, recent studies have pointed out that several limitations exist in the current LMD architectures [22].

In particular, this work addresses the following Research Gaps (RGs) related to the application of advanced technologies in Last-Mile Logistics:**RG1.** Lack of interoperable LMD frameworks including both AI algorithms and user-centered interfaces for situational awareness and operational assistance [6];**RG2.** Limited support for context-aware route planning and real-time decision-making in dynamic environments [9];**RG3.** Simplistic usage of LLMs, lacking integration of KG and edge computing technologies [23].

Building on these considerations, this work introduces the following contributions:A reference LMD framework for AI-based fleet management, integrating a load optimization algorithm, formulated as a specialized Capacitated Vehicle Routing Problem (CVRP) and tailored to LMD constraints;A route-planning module extending a general-purpose vehicle routing engine with customizable AI-based traffic prediction models, which leverage data from urban sensor networks, such as traffic flow sensors and vehicle telemetry systems. It aims to dynamically reorder delivery points to respond to real-time conditions, reducing delays and operational costs;An MR-assisted module for HMDs equipped with advanced micro-devices including a camera array, depth sensors, and Inertial Measurement Units (IMUs) to support warehouse operators during loading tasks. HMDs guide workers through visual overlays, increasing speed and accuracy in delivery operations;A KG-driven LLM conversational assistant to optimize Last-Mile Logistics. The KG provides structured and semantically rich data (e.g., vehicle status, warehouse information), whereas the LLM combines user queries with contextual data to assist logistics operators in offering a simple walkthrough in specific tasks;A cloud-based prototype implementation extending open-source technologies and tools with domain-specific enhancements;A case study regarding LMD operations in the Metropolitan City of Bari and the Apulia region, highlighting key value propositions of the approach.

The system is implemented as a platform composed of several microservices, leveraging web technologies to enable bidirectional real-time coordination between fleet managers and warehouse operators. This modular approach allows the different logical components to be decoupled and deployed as individual containerized services, improving the overall maintainability and scalability of the system, as is discussed in Section 3. The KG acts as a central repository for semantic data management, while the RAG-based conversational assistant provides an intelligent natural interface for querying the system’s status, retrieving relevant knowledge, and executing operational commands. Through MR visualization, warehouse operators can track shipment allocations dynamically and interact with the backend system without manual data entry, reducing errors in package handling and delivery. To assess the practical applicability of the proposed framework, a case study is presented for illustrating its deployment in a realistic Last-Mile Delivery scenario. The case study demonstrates how the integrated system supports fleet optimization, knowledge-based decision support, and interactive MR interfaces in a dynamic logistics environment. By showcasing the interaction between warehouse operators, fleet managers, and AI-driven components, the case study provides concrete insights into the system’s functionality and potential impact on operational efficiency. The remainder of the paper is as follows:Section 2 discusses related work and provides a comparison between the proposed framework and the current state of the art;The proposed architecture is reported in Section 3, along with a detailed description of data management strategies (Section 3.1), route planning and optimization algorithms (Section 3.2), AI-based conversational assistant (Section 3.3), and mixed-reality interfaces (Section 3.4);Section 4 includes a case study (Section 4.1) highlighting the peculiarities of the framework in a realistic setting in the Apulia region. This section also discusses the results of the experimental evaluation of the RAG pipeline (Section 4.2) and finally provides a discussion of benefits and possible limitations of the proposed approach with respect to the aforementioned RGs (Section 4.3);Section 5 reports on the conclusions of the study and outlines potential directions for future work.

## 2. Related Work

Last-Mile Logistics (LML) is a quickly evolving topic, which is shaped by trends in consumer behavior [24] and by new concepts introduced by the day. In the era of parcel lockers [25], delivery drones [26], electric mobility [27], and intelligent warehouses [22], and with huge industrial interests, research chases practice more often than it guides innovation, thus returning fragmented pictures [28]. Implications for novel AI-based proposals to support LMD are equally multifaceted. This work focuses on an architectural and technological perspective.

The importance of increasing LMD resilience and adaptability through big data sensing and descriptive, predictive, and prescriptive analytics models cannot be overestimated [29]. Several works have focused on solving Vehicle Routing Problems (VRPs) using exact, heuristic, and metaheuristic algorithms [30] in order to optimize the distribution of goods to customers via a planned route and achieve the shortest distance under specific constraints, such as minimum cost and shortest time. Recent advancements have seen the integration of Machine Learning (ML) algorithms to enhance the accuracy and adaptability of route planning. Reinforcement learning and deep neural network models have shown promising results in learning optimal delivery routes based on historical data, real-time traffic conditions, and other relevant factors [31]. Nevertheless, models based on Artificial Neural Networks (ANNs) often require computationally expensive training on specific large-scale datasets to achieve optimal results [32], making it difficult to generalize these techniques and limiting the ease of application to a wide range of real-world logistics scenarios [33]. Recent approaches have explored distributed intelligence models for the real-time processing of operational data, improving adaptability in dynamic logistics scenarios [34]. Similarly, a serverless microservice architecture proposed in [35] demonstrates how AIs-driven edge computing can support large-scale sensor network applications, with potential benefits for operations in sensorized logistic facilities, including decreased latency and enhanced responsiveness.

With the rise of Augmented Reality (AR) and MR applications in logistics, the interaction between AI-driven approaches and human workers has also been explored to improve the efficiency of LMD operations, optimizing tasks such as package picking, sorting, loading, and unloading. Virtual Reality (VR) environments are used as simulation tools to assess the impact of different fleet configurations, route planning algorithms, and scheduling policies [36], whereas AR technologies are usually applied to streamline package processing in distribution centers [37]. By using AR on mobile devices [38] or headsets, workers can receive real-time visual guidance on optimal loading sequences, resource placement, and sorting priorities. This not only improves efficiency but also reduces the likelihood of errors during several phases of Last-Mile Delivery. Recently, MR tools have been increasingly integrated in commercial delivery solutions. For example, Amazon Transportation announced the implementation of a Vision-Assisted Package Retrieval (VAPR) system to support operators during package delivery, providing multi-modal feedback [39]. Nevertheless, as summarized in Table 1, despite significant advancements in both optimization algorithms for Last-Mile Logistics and AR/MR technologies, a relevant gap exists in their integration. As reported in Section 1, this limitation represents an important aspect in current research and industrial proposals. Another open issue is the limited support for context-aware route planning and decision-making in dynamic environments.

In recent years, several works have explored LLM-based applications to support logistics through real-time decision support, route optimization, and operator assistance. As described in [42,43], AI chatbots represent a relevant tool in enhancing customer service for Last-Mile Logistics, offering automated handling of delivery status inquiries and rescheduling requests. A RAG framework is presented in [17], integrated into a digital twin architecture originally designed for energy infrastructure management, but also applicable to logistics. The framework integrates structured domain knowledge with real-time query resolution, allowing operators to obtain context-aware responses to challenges in resource monitoring and management. While this enhances decision support, the study highlights challenges related to integration complexity, domain adaptation, and security constraints when deploying AI-driven knowledge retrieval systems in operational settings. In [40], a LLM agent is employed to enhance LMD routes by creating optimized delivery sequences according to pre-established constraints. A significant reduction in total delivery distance is observed, demonstrating the capability of LLMs in logistics optimization. However, the reported low accuracy and the lack of real-time traffic awareness constrain the real-world usage for dynamic logistics scenarios. A capability-based AI framework for logistics is also introduced in [44], mapping LLM functionalities to supply chain decision-making processes. The framework identifies AI-driven interaction, prediction, and reasoning as core components in logistics automation, particularly in fleet management and warehouse operations. However, the study remains at a very high-level, providing strategic guidance rather than specific technological solutions. Similarly, the work in [41] demonstrates how LLMs can support operators lacking specialized expertise in analyzing information related to specific machine tasks through conversational interfaces.

While previous studies have explored LLM-based assistants for logistics, these solutions primarily focus either on route optimization or customer service automation without direct integration with MR interfaces. Existing approaches often operate in isolation, utilizing historical data, demand forecasting, and standalone conversational assistants, but fail to leverage MR to enhance real-time adaptability and situational awareness. A major limitation of current research is the lack of interactive Decision Support Systems (DSSs) that combine sensor-enabled edge computing technologies and MR visualization with advanced LLM-based reasoning. Dynamic factors such as real-time delivery constraints, unexpected route changes, and contextual worker assistance remain largely unaddressed in existing solutions. The absence of integrated AI-driven MR interactions prevents logistics operators from fully utilizing immersive, voice-driven chatbots for real-time navigation, decision-making, and task optimization.

To overcome these limitations, the proposed approach introduces a novel synergy between MR interfaces and LLM-powered chatbots by integrating real-time bidirectional communication between operators, knowledge graphs, and AI assistants through MR-enhanced overlays. The proposed system enables interactive and contextual guidance for delivery personnel, warehouse operators, and fleet managers. This vision introduces a significant novelty in achieving a real adaptive delivery ecosystem, bridging the gap between AI-driven conversational models and immersive MR-based logistics support.

## 3. Framework Architecture

The proposed framework, sketched in Figure 1, is designed as a REpresentational State Transfer (REST) multitenant microservice architecture. The back-end exposes functionalities such as configuring the initial state of the fleet scheduler, starting and stopping the optimization process, retrieving the schedule of the fleet, etc., through a facade REST API.

Before introducing the clients, it is important to note that the system currently supports users with two distinct roles:*Managers*, who have full access to the system. They are able to edit the fleet and shipments configuration, access order detail information, and start/stop the optimization process.*Warehouse operators*, endowed with limited read access to the fleet and shipment schedule and capable of updating vehicle status by performing loading operations on physical parcels.

Clients include a mixed-reality application for the Microsoft (Redmond, WA, USA) HoloLens 2 (https://www.microsoft.com/hololens, accessed on 4 November 2024) headset platform, which enables warehouse operators to receive real-time information about the allocation of each parcel to specific vehicles and to mark parcels as loaded into the appropriate vehicle through easily accessible MR controls. The application also incorporates a conversational assistant component, enabling hands-free interaction for features such as marking orders as loaded onto vehicles and indicating when products have been moved between shelves. Additionally, the MR client may also be used by warehouse managers, who have access to more detailed information, such as customer and delivery details, and benefit from additional functionalities, including the ability to reschedule orders onto different vehicles.

A second web-based client allows managers to configure the system, setting parameters such as delivery locations, the distribution of the fleet among distribution hubs, the capacity of each vehicle, the daily departure time, and more. Following the initial setup, the optimization process can be initiated via this interface, which is then refreshed periodically to present the best solution available at each given moment. The manager can then wait until the solver halts, according to the criteria described in Section 3.2, or manually stop the Optimizer if the solution is deemed to be satisfactory. A read-only, limited information variant of the interface can finally be displayed in mixed reality by warehouse operators, allowing them to see further information about the current state of the fleet and shipments.

The proposed architecture is designed to be highly modular and extensible. Specifically, it supports the integration of additional warehouse automation technologies, including Automated Storage and Retrieval Systems (ASRSs), Automated Guided Vehicles (AGVs), Autonomous Mobile Robots (AMRs), and cobots, without requiring significant changes to the underlying framework. Extension with additional microservices and Application Programming Interfaces (APIs) for Machine-to-Machine (M2M) communication and agent-based hyperautomation leverages the machine-understandable KG representing warehouse knowledge. This enables a seamless addition of autonomous components capable of performing order picking and placement tasks and of interacting with human staff.

The 4 + 1 architectural view model in Figure 2 looks at the proposal from multiple perspectives to clarify its structure and operational dynamics. The *logical view* highlights the main software components, while the *development view* illustrates how these components are organized into independent microservices, as depicted in Figure 1. The *process view* focuses on runtime interactions among services, particularly those involved in semantic data management (Section 3.1), optimization (Section 3.2), and human–AI communication (Section 3.3). The *physical view* presents the deployment of containerized services within a cloud-native infrastructure (Section 4). Finally, the *scenarios view* captures the main user interactions as described in Section 4.1. This multi-layered view offers an extensive insight into the system’s architecture and function across various technical and operational aspects.

### 3.1. Semantic-Aware Data Management

The Knowledge Graph (KG) component serves as the shared semantic data model underpinning the whole framework, providing a structured and interoperable representation of crucial entities such as fleet configuration, order details, and warehouse organization, including shelves, products, and storage conditions. By acting as a central repository, it facilitates a seamless data exchange between core system components, including the Optimizer and the AI-driven chatbot interface. Knowledge-based interoperability also promotes collaboration with external supply chain actors—e.g., cargo logistics or reverse logistics providers—and loosely-coupled coordination across multiple warehouses in a large area [45]. Unlike traditional relational or document-based storage approaches, the KG leverages a graph-based ontology model, offering a flexible and semantically enriched way to manage complex relationships and dynamic updates.

To achieve high-performance knowledge representation and querying, the KG is implemented using *Cowl* version 0.7.2 [46], a lightweight library optimized for handling Web Ontology Language (OWL) [47] ontologies. Cowl was selected by virtue of its high time performance even when processing KGs containing millions of axioms [46,48], while still maintaining a significantly lower memory consumption compared to other OWL manipulation tools. Using an OWL-based KG enables the formal encoding of domain knowledge with well-defined semantics, ensuring consistency and extensibility in data representation. The adoption of OWL provides several advantages: it allows for an expressive schema definition, supports semantic interoperability across system components, and enables sophisticated query mechanisms beyond simple key–value retrieval. Furthermore, OWL facilitates the enforcement of logical constraints and the structuring of domain concepts into class hierarchies, making it an ideal choice for managing the typical relationships inherent in warehouse and fleet operations.

The reference ontology models entities including vehicles, shipments, products, warehouse shelves, and sectors. Figure 3 depicts the main classes and properties in the graph-based data model, with their relationships: OWL classes are blue ovals and XML Schema Datatypes (XSD) [49] are green rectangles; OWL object properties are blue arrows while datatype properties are green arrows. Mentioned entities are described in Table 2.

A RESTful API to access the KG grants interoperability and modular integration within the broader system. Through standard API endpoints, other components can perform both read and update operations efficiently, ensuring that fleet assignments, warehouse stock levels, and order statuses remain synchronized across all interfaces. Additionally, the KG supports serialization, allowing for its current state to be saved to persistent storage and restored as needed, enabling fault tolerance and long-term data retention without loss of information.

A notable feature of the OWL-based KG is the potential to leverage logic-based reasoning to derive implicit knowledge from explicitly stored assertions, introducing inference capability at the retrieval level. A simplified ad hoc inference engine is integrated to resolve class hierarchies and relate them to instance-level assertions, thereby augmenting knowledge retrieval processes. For instance, if different classes of delivery vans are defined as subclasses of *Vehicle* within the KG—such as *Light_Van*, *Medium_Van*, and *Heavy_Van*—a logical inference can automatically associate new vehicle instances with the correct class based on their load capacity. If a new van model is introduced with a specified maximum weight and volume, the system can infer its classification and apply the appropriate operational constraints without requiring manual intervention. This inference-based augmentation capability enhances the flexibility of the system, allowing for more robust automation in fleet and warehouse management while reducing the need for constant manual rule updates.

### 3.2. Route Planning and Optimization

The Optimizer component of the framework is built on top of *OptaPlanner* version 9.38.0 (https://optaplanner.org, accessed on 4 November 2024), an open-source, AI-based optimization engine designed to address complex constraint satisfaction and optimization problems in operations research areas such as employee rostering, task scheduling, and supply chain optimization. It leverages advanced algorithmic techniques, including heuristics, local search metaheuristics (Tabu Search [50], late acceptance [51], etc.), and evolutionary algorithms to find optimal solutions under a set of constraints. While algorithmic approaches to optimization are known to be computationally intensive, OptaPlanner is a mature tool that has demonstrated robust scalability in practical applications. As an example, in [52], it was successfully used to solve vehicle routing problems involving up to 500 customer nodes within 2–3 min on standard commodity CPUs, underscoring its efficiency and suitability for real-world deployment scenarios.

For the proposed framework, the solver was customized to optimize a CVRP through the application of a First Fit Decreasing construction heuristic to determine an initial solution, and then iterating via Late Acceptance local search until either a global timeout is reached or the solution stops improving for a sufficiently long time. Both timeouts are configurable by the manager. In a CVRP, the Optimizer must provide an allocation of shipments to vehicles in the fleet such that the overall delivery time is minimized under the following constraints:The allocation must never exceed the capacity of each vehicle, which can be specified in terms of both a maximum volume and weight;The overall travel time of each vehicle must not exceed a maximum threshold, which can be configured to account for driver shift times;All parcels must be delivered (optional).

The last constraint may make the problem unsolvable, as there may be situations in which it is impossible to satisfy all constraints (e.g., the shipments exceed the cumulative capacity of the fleet). If that is the case, lifting it allows the system to still provide an optimal schedule, albeit at the cost of excluding some parcels from the current delivery round.

As shown in the sequence diagram in Figure 4, a manager starts the optimization process via the web-based client. While the Optimizer runs, it first retrieves the current state of the system, stored in the KG component. Then, it queries the Route Planner component to estimate the time it would take a vehicle to travel from one waypoint to the next. For each waypoint pair, the Route Planner computes the best route by means of *GraphHopper* version 7.0 (https://graphhopper.com, accessed on 4 November 2024), an open-source routing engine designed for fast and efficient pathfinding and navigation across road networks and other types of graphs, and which was used in LMD for problems of a realistic scale, e.g., with 20 vehicles and 1000 delivery stops in [53]. In addition to being used by the Optimizer to obtain an initial estimate of the travel time, the computed route is shown in the web application and may be integrated into additional clients for use by, e.g., couriers and other logistics operators.

The estimated time for the route is subsequently adjusted by means of the Traffic Predictor, a component that can leverage historical traffic data to refine the time estimation provided by GraphHopper. This module has a generic interface whose inputs are a route, i.e., a sequence of geo-localized edges between two waypoints, and a timestamp, representing the moment the vehicle enters the specified route. The predictor is expected to output an estimated travel time for the route or nothing if, e.g., it does not have data for that specific route, in which case the Route Planner falls back to GraphHopper’s initial estimate. This general interface allows the underlying implementation to use a wide array of prediction methods, ranging from statistical analysis (time series analysis, regression models, etc.) to more sophisticated ML approaches. The computed routes and their estimated travel time are used to align the internal status of the KG and to return the best solution available at the moment.

A prototype implementation of the Traffic Predictor interface was developed based on a model trained using data provided by the *TIM City Forecast* (https://www.timenterprise.it/5g-e-iot/data-analytics/tim-city-forecast, accessed on 4 November 2024) platform. The dataset contains over 8 million Origin-Destination (OD) flow records estimated by grabbing the activity of mobile network Subscriber Identity Module (SIM) connections anonymously every hour over a 12-month period, focusing only on nearby cell towers. Movement patterns between the city of Bari and ten surrounding municipalities are represented. Each data point corresponds to a temporal OD flow, with additional metadata such as time and day retained to support contextual analysis. A traffic prediction model was trained on this dataset based on a Convolutional Neural Network (CNN) architecture tailored for time series prediction [54], with a horizon of up to 12 h. The model processes time series of historical traffic counts, structured as sequences of OD flow values, and employs convolution layers to capture local temporal dependencies and short-term trends. The input of the network is further enriched with temporal context data and holiday information, allowing it to implicitly extract recurring fluctuations in traffic volumes associated with daily and weekly routines, like rush hours and weekend effects.

The whole process described so far is executed for each iteration of the solver, making the optimization of its runtime imperative to achieve a sufficiently refined solution within an acceptable time frame. As one might expect, route computation and traffic estimation dominate the overall runtime. To minimize the first component, GraphHopper is configured to use Contraction Hierarchies [55], a technique that introduces precomputed “shortcuts” in the overall graph, greatly reducing the search space and, thus, leading to a significantly shorter computation time for each query. Furthermore, since queries to the Route Planner are always directed to a specific set of *N* start and end coordinates (i.e., warehouses and delivery locations), an associative cache is introduced with waypoint pairs as keys and routes as values. To increase the performance of the traffic estimation step, the following strategy was adopted: if route entry timestamps are quantized to some time duration *T* (e.g., hourly), the Traffic Predictor can be invoked with at most $M=\lceil T_{last}/T\rceil$ different inputs for each route, with $T_{last}$ being the last vehicle reentry time at its respective warehouse. The cache can, thus, be extended with a further dimension, storing travel times for each route and entry timestamp. In this configuration, the cache has a worst-case size of $M\cdot N^{2}$ entries, which may be an acceptable space tradoff if *N* is not too large (e.g., for $N<10^{3}$). The introduction of the cache has a significant impact on the runtime of each solver iteration, making it possible to move from less than 10 steps/s to about 2000 steps/s on the testbed device (2021 MacBook Pro, M1 Max SoC, 64 GiB RAM), with a ∼200-fold decrease in step computation time.

### 3.3. AI-Based Conversational Interface

The Conversational Assistant component of the framework, depicted in Figure 5, serves as an intelligent conversational agent, enabling seamless interaction with warehouse and fleet management roles. Designed for hands-free operation, it provides an efficient means for operators and managers to query system data, execute operational commands, and receive contextualized responses. At its core, the conversational interface is powered by a RAG pipeline that integrates multiple AI-driven components to ensure accurate and context-aware responses.

The RAG pipeline is built upon a combination of state-of-the-art technologies. The inference engine is implemented using *llama.cpp* (https://github.com/ggml-org/llama.cpp, accessed on 4 November 2024), a lightweight and highly optimized C++ LLM inference library supporting a sizable selection of models, including the *LLaMA* [56] and *Mistral* [57] model families. By leveraging quantization techniques, llama.cpp enables low-latency inference on consumer hardware (both CPU and GPU) at a fraction of the computational cost typically required by traditional deep learning frameworks. Semantic search capabilities are provided through *Faiss* [58], a high-performance vector index developed by Meta’s Fundamental AI Research. FAISS is optimized for rapid and scalable nearest-neighbor searches in high-dimensional spaces, making it particularly suitable for large-scale semantic similarity tasks. The embedding model utilized for knowledge representation is *SBERT* [59] (*all-minilm-l6-v2*), a Sentence-BERT variant optimized for fast and efficient sentence-level embedding generation. SBERT encodes textual descriptions into dense vector representations, capturing semantic similarities between entities, and is particularly well-suited for tasks such as semantic search, clustering, and retrieval-based question answering. These components work together to support intelligent query handling and decision-making within the RAG framework.

Upon loading or updating the KG, the system pre-processes the semantic data by computing SBERT embeddings of all referenced OWL entities and storing them in the FAISS vector index. When a user submits a query, an embedding of the query text is computed and compared against stored embeddings to retrieve the top-K most relevant entities based on cosine similarity. These steps are illustrated in Figure 6, showing an example of how a warehouse operator can interact with the conversational assistant through the MR client.

Once relevant entities are identified, the KG is queried to extract all axioms referencing these entities. The query mechanism can be configured to traverse the graph up to a specified depth, enabling contextual expansion of the retrieved knowledge. The resulting axioms are transformed into a structured JavaScript Object Notation (JSON) representation, which is then embedded into the LLM prompt. This structured knowledge integration ensures that the chatbot operates with up-to-date and contextually rich data, enhancing both the accuracy and interpretability of responses.

At the core of the decision-making process, the LLM model is tasked with determining the appropriate response to the user query by running two inferences: the first to determine whether to generate a conversational reply or invoke a predefined function, such as loading an order onto a vehicle, rescheduling a shipment, or marking a product as moved between shelves. If a function is called, its output is incorporated into a subsequent prompt, and the model is inferenced again to generate a final response that clearly explains the action taken and its outcome. If no function is required, the model undergoes a second inference to generate a conversational response directly.

To enable hands-free operation, the system integrates a voice User Interface (UI) implemented through *whisper.cpp* (https://github.com/ggml-org/whisper.cpp, accessed on 4 November 2024), an optimized C++ implementation of OpenAI’s *Whisper* [60] model for Automatic Speech Recognition (ASR). Whisper is a multi-lingual, general-purpose speech recognition model trained on a large-scale dataset, capable of transcribing spoken language with high accuracy. The conversational assistant utilizes a compact variant of Whisper to balance performance and computational efficiency, ensuring smooth real-time transcription in warehouse environments where operators require quick hands-free interactions. Users can activate the chatbot using a predefined trigger word, which is processed on-device (see Section 3.4), after which their speech is recorded and then sent to the backend for ASR through Whisper and further processed as a standard textual query. This feature allows warehouse operators and managers to interact with the system without the need for manual input, significantly improving usability in environments where hands-free operation is critical.

By combining efficient knowledge retrieval, structured reasoning, and a flexible multimodal interface, the AI-based conversational agent enhances situational awareness and operational efficiency. Its integration into the overall framework provides an intuitive mechanism for querying warehouse and fleet status, managing shipments, and executing operational commands, ultimately streamlining LMD logistics.

### 3.4. Mixed Reality for Efficient Operations

The framework integrates a MR client warehouse operators can leverage to display real-time information on the current state of the fleet and parcels. The client exploits the Quick Response (QR) code detection capabilities included in the Mixed Reality Toolkit (MRTK) API running on the Microsoft HoloLens 2 headset to provide the operator with contextual information about each parcel, as shown in Figure 7. Once a parcel has been loaded onto the designated vehicle, the operator can push a virtual button to mark it as loaded. This event triggers a KG update, making the new data available to all other clients so that the warehouse personnel is always provided with up-to-date information.

The MR client was developed as a *Unity* (https://unity.com, accessed on 4 November 2024) application using Microsoft MRTK (https://learn.microsoft.com/windows/mixed-reality/mrtk-unity, accessed on 4 November 2024) version 2. Unity is a comprehensive and versatile real-time development platform widely used for creating 2D, 3D, VR, and AR content. The MRTK is an open-source project designed to accelerate the development of mixed reality applications across a wide range of devices, including Microsoft HoloLens, Windows Mixed Reality headsets, and other MR platforms. It provides a comprehensive collection of scripts and components aimed at simplifying the implementation of spatial interactions, user interfaces, and realistic physics in a 3D environment. The foundation of the whole software stack is *OpenXR* (https://www.khronos.org/openxr, accessed on 4 November 2024), an open, royalty-free standard designed to provide a common set of features for MR platforms, abstracting away the hardware-specific implementations.

Specifically, the client exploits Microsoft’s QR (https://www.nuget.org/packages/Microsoft.MixedReality.QR, accessed on 4 November 2024) Software Development Kit (SDK) to gain access to QR codes detected in the scene by the HoloLens 2’s built-in vision driver, and it uses the spatial information provided by the SDK to anchor contextual UIs to the parcels. Each QR code carries a unique identifier for the parcel, which serves as a key to fetch pertinent information via a REST API call to the KG component. This process seamlessly updates the MR interface with the latest data, ensuring warehouse operators have access to real-time information. The parcel identifier is also used in other API calls, such as when displaying additional information through the web-based interface, and when updating the load state of the parcel.

To further enhance operational efficiency, the MR client shown in Figure 7 integrates a conversational interface that provides real-time assistance and facilitates hands-free interaction. The assistant is persistently displayed in the bottom-right corner of the MR interface, ensuring immediate accessibility without obstructing critical operational views. This interface displays the ongoing exchanges between the user and the chatbot in a textual format, allowing operators to track conversation history and reference past interactions. By embedding this interactive assistant into the MR interface, working as explained in Section 3.3, warehouse operators gain an intuitive and efficient tool to streamline their workflow, reducing reliance on manual input and minimizing operational disruptions.

## 4. Evaluation

This section reports on an evaluation of the proposed framework from both practical and technical perspectives. First of all, a prototypical implementation is presented, demonstrating the integration of AI-based fleet optimization, MR interfaces, and a KG-driven conversational assistant in a LMD scenario in the Apulia region. The discussed deployment highlights the framework’s capability to support warehouse operations and fleet management in real-world conditions. Subsequently, an experimental assessment of the RAG pipeline is discussed, focusing on its effectiveness in knowledge retrieval, graph management, and response-generation tasks. The results provide insights into the trade-offs between a language model’s complexity, execution time, and accuracy in the integration of LLMs for conversational task- and resource-oriented assistants. Finally, a discussion in the perspective of Research Gaps outlined in Section 1 is proposed.

The setup used for the prototype implementation is shown in Figure 8, materializing the physical standpoint of the 4 + 1 architectural model depicted in Figure 2. Specifically, core services (such as the Optimizer, KG, and conversational assistant) can be deployed as containerized microservices in cloud or on-premise settings. Client applications, including the web dashboard and the HoloLens 2 MR interface, may interact with these internal services through a public RESTful Gateway API, which handles request routing, service resolution, and client authentication and authorization. The microservices can be executed on a single Virtual Machine (VM) instance, such as an *Amazon EC2* (https://aws.amazon.com/ec2/, accessed on 4 November 2024) node, or scaled and orchestrated using platforms like *Kubernetes* (https://kubernetes.io/, accessed on 4 November 2024) or *Amazon Elastic Container Service (ECS)* (https://aws.amazon.com/ecs/, accessed on 4 November 2024), depending on workload requirements. This flexibility enables the framework to support scalable, modular, and production-ready system deployments in LMD scenarios.

### 4.1. Prototype Deployment in a Realistic Scenario

Let us consider the typical Last-Mile Delivery scenario, where a company aims to establish the most efficient schedule for a fleet of vehicles to deliver a defined quantity of items to various destinations. In our specific case study, the problem instance involves a fleet of 10 vehicles, a single warehouse, and a schedule encompassing 100 shipments that need to be fulfilled. The framework provides the capability to simulate a problem by employing a pseudo-random synthetic data generation method to instantiate variable values dynamically. Table 3 reports the range of values for the relevant parameters involved in the process. Additionally, the geographical location of each shipment is randomly generated within a bounding shape delimited by a polyline, delineating the confines of the Apulia region. Alternatively, as described in Section 3, the manager can set all parameters that characterize the specific problem instance. This facilitates the (re)creation of realistic scenarios, enabling users to align the problem definition closely with real-world situations and requirements.

Upon defining the problem instance, the manager can initiate the optimization procedure by interacting with the web application illustrated in Figure 9 and clicking on the “Start” button (box A). The highlighted routes on the map (box B) are determined by the Optimizer, which is able to compute the optimal scheduling for the vehicles in the fleet, accounting for the constraints specified in Section 3.2. The map is periodically updated with what is currently found to be the best solution. Hovering the mouse cursor over map markers allows users to access information regarding the respective shipments. This includes details such as weight, volume, reference vehicle, and status (whether loaded or still in the warehouse). The Optimizer systematically iterates to enhance the existing solution (score, total time, and cumulative travel distance, as shown in box A) until either a halt condition is reached, or the manager manually stops the solver. In either case, the procedure can be resumed by clicking on the “Start” button, in which case the Optimizer keeps iterating from the state that is currently displayed. Box C in Figure 9 contains the final optimized schedule for the vehicles, providing comprehensive details for each vehicle, including the total load volume and weight, as well as the distance and time required to fulfill the respective deliveries.

The warehouse operator, equipped with a HoloLens 2 headset, interacts with a MR interface consisting of several panels, as depicted in Figure 7.

Proximity Menu (box A): notifies the operator when the optimization process is finished, offering details about the cumulative number of parcels loaded onto the available vehicles. The panel seamlessly tracks the operator’s movements and can be anchored to a specific point in space as needed.Parcel Information Panel (box B): triggers upon detecting the parcel’s QR code and provides details about the estimated delivery date-time, assigned vehicle, and loading status.Parcel Picking Button (box C): upon loading the tracked parcel onto the designated vehicle, the operator can communicate this status update to the system by pressing the corresponding virtual button. This information is propagated in real time to all web application and MR clients.Parcel Detail Button (box D): provides additional details about shipments and assigned vehicles through an interactive web interface. This view can be materialized in the MR scene by pressing the “View on map” button, which triggers the appearance of a web view containing a read-only version of the interface shown in Figure 9, with a focus on the delivery location of the specific parcel.Conversational Assistant Panel (box E): facilitates hands-free access to essential warehouse data and operational functions for operators. Through voice interaction, it enables users to retrieve an inventory’s status, locate items, verify order details, and execute predefined warehouse tasks, as specified in Section 3.3 and Section 3.4. This capability enhances operational efficiency while minimizing dependence on handheld interfaces.

The MR interface updates in real time, tracking the parcel’s position as it is manipulated, and global information is always kept up to date and displayed in the proximity menu. Furthermore, employing a voice-activated assistant provides several advantages, including hands-free operations, faster task completion, and increased safety by reducing the need for physical interaction with devices. Together, these benefits ensure all operators have access to the latest information, potentially contributing to a reduction in errors and enhancing the overall operational efficiency and workforce safety.

Although the evaluation focuses on LMD operations, the integration of a machine-understandable KG based on a common conceptual vocabulary allows the proposed framework to be adapted for multi-level logistics scenarios, including first-mile pickups and middle-mile hub coordination. In particular, the graph can be enriched with concepts and relationships for earlier-stage logistics actors—such as suppliers, distribution centers, and regional hubs—enabling the KG to support data integration across multiple operators and logistics stages. For example, a parcel’s lifecycle can be represented as a sequence of annotated events that can be consumed by both human-facing interfaces and autonomous components. Let us consider a middle-mile logistics partner that shares information about a batch of shipments by annotating relevant metadata (e.g., estimated arrival time, route origin, and handling instructions) and publishing it to the LMD KG RESTful API. In this way, LMD warehouse operators can visualize incoming deliveries on the HoloLens 2 device and proactively schedule loading operations.

While the case study focuses on the Apulia region, the proposed framework is designed with generalization in mind. The adopted methods and technologies (e.g., the use of KGs and RAG) are scalable by design—as assessed in Section 4.2—and well suited to support logistics operators globally, representing a key innovation aspect of the approach. The core system components, such as the optimization engine, the MR interface, and the KG-driven assistant, are agnostic to geographical location and can be deployed in other urban or regional settings with minimal adjustments. Specifically, domain-specific aspects of warehouse operations, such as layout and item handling rules, are managed through the KG, granting inherent adaptability to different logistics contexts. Geographic portability primarily requires updating the underlying cartographic and routing data to reflect the new area, which can be easily performed through the *OpenStreetMap* (https://www.openstreetmap.org, accessed on 4 November 2024) data export web interface. On the other hand, the prototypical Traffic Predictor described in Section 3.2 is trained on mobility patterns specific to the Bari metropolitan area and would need to be retrained. However, it is worth noting that it is integrated as an optional component in the time estimation pipeline, falling back to GraphHopper’s travel estimates if it is missing or if data are not available for specific locations. It is also important to note that the Traffic Predictor is abstracted behind a general interface, allowing the underlying model to be trained on a wide range of data sources, including public telemetry datasets, real-time or historical data collected from road sensor networks [61], or delivery records maintained by logistics providers themselves. Furthermore, the predictor interface is model-agnostic: its internal implementation can range from simple statistical regressors to more complex machine learning or deep learning architectures, depending on the availability of data and computational resources in the target deployment setting. The overall architecture is, thus, well suited for diverse logistics scenarios owing to its modular design and decoupling of components from specific data sources or models.

### 4.2. Experimental Assessment of the RAG Pipeline

To systematically assess the performance and accuracy of the RAG pipeline in retrieving warehouse-related information and executing configuration modifications via function calls, a test set consisting of 45 distinct queries was designed. These queries are distinguished in two primary categories: (i) information retrieval queries that do not require function invocation; (ii) action-oriented queries needing specific function calls to modify warehouse configurations.

The first category includes queries aimed at retrieving structured information, such as requesting details about a specific product, accessing comprehensive shipment information (e.g., delivery address, estimated delivery date, buyer details), and obtaining an overview of the warehouse status, including available vehicles, warehouse sectors, and shelf organization. These queries were employed to evaluate the system’s ability to extract relevant information from the KG without the need for external function execution. The second category consists of queries requiring function invocation to modify warehouse operations, such as (i) moving a specified quantity of products of the same type from a source to a target shelf; (ii) loading a specific shipment order into the vehicle scheduled for its delivery; (iii) rescheduling a shipment by reassigning it to a different vehicle. These queries were designed to test the system’s capability to accurately determine when function execution is required, select the appropriate function, and supply the correct parameters.

The evaluation was carried out on the same testbed device mentioned in Section 3.2, using four distinct LLM models, with their main characteristics outlined in Table 4, to examine performance differences across various architectures. The assessment was structured around two key dimensions: task-specific performance metrics and correctness evaluation. Processing times, measured in milliseconds and displayed in Figure 10, reflect the mean computed across all queries. The results include the following performance metrics:Function selection, which measures the time taken by the LLM to determine the appropriate function (if needed) after receiving the prompt containing the necessary knowledge;Response, which measures the time needed to generate a conversational response to the user;Overall, representing the total time required for query embedding, retrieval from the knowledge graph, function selection, function execution, and response generation.

Note that query embedding, retrieval, and function execution are independent of the LLM model, and their computational cost is negligible compared to the model inference (<30 ms overall, on average). Therefore, their bins are not included in the plot in Figure 10.

The observed processing times exhibit a clear dependence on the complexity of the underlying models, as expected. More advanced models, such as Llama 3.3 (70B) and Mistral Small 3.1 (24B), with a higher number of parameters and more complex architectures, display longer processing times across all performance metrics when compared to smaller models. The Function selection time is influenced by the model size, with larger models requiring additional time to process the input and to retrieve the relevant knowledge for accurate decision-making. The Response time is similarly affected by the model’s scale, but it is almost an order of magnitude lower due to caching mechanisms. In transformer-based LLMs, caching stores intermediate key and value tensors from the self-attention layers, allowing for previously computed representations to be reused instead of recalculating them. Since the prompt used for function selection shares the same prefix as the one used to generate the final response, including embedded knowledge retrieved from the KG, cached activations from the first inference can be applied directly to the second. This reduces redundant computation, leading to a substantial decrease in response time. These findings confirm that processing times are inherently tied to model complexity, aligning with theoretical expectations regarding the architecture and functionality of state-of-the-art LLMs.

The correctness evaluation, as shown in Figure 11, was performed through multiple validation layers, with each layer assigned a corresponding rating in the range of [0.0,1.0]:Function name: a binary score that indicates whether the correct function has been selected for execution. A score of 1 means the correct function has been chosen, while a score of 0 represents an incorrect function selection;Function args: a metric that quantifies the correctness of the parameters provided to the selected function. It is computed as$$function\_args\_score=\frac{\left(p_{1}+p_{2}+\cdots +p_{N}\right)}{N}$$
where $p_{1},\ldots ,p_{N}$ represent the scores assigned to each argument, and *N* is the total number of function parameters. For atomic arguments such as strings and numbers, $p_{i}=1$ if the argument is correct, and $p_{i}=0$ otherwise. In case of collection parameters, such as arrays and dictionaries, $p_{i}$ is obtained as$$p_{i}=\frac{\#correct}{\#correct+\#missing+\#incorrect}$$
where #correct, #missing, and #incorrect denote the number of correctly provided, missing, and incorrect elements within the collection, respectively.Output (embed): the cosine similarity between the embeddings of the generated response and the expected output. This metric reflects how semantically similar the provided response is to the expected output, with higher values indicating greater alignment;Output (LLM): a score obtained by leveraging a “judge” language model (Llama 3.3 70B) [62]. The judge is instructed to assess the correctness of the response relative to the expected value through a few-shot prompt-engineering strategy. The model is prompted to generate a quantitative score, evaluating both the logical consistency and factual accuracy of the output and disregarding any grammatical and formal differences;Output (human): a score assigned through manual evaluation, where a human assesses how well the generated response aligns with the expected one. This evaluation quantifies the correctness and reliability of the output based on a direct comparison with the reference response.

Results in Figure 11 represent the average correctness values for each score, computed across all 45 queries for each assessed LLM model. It is important to highlight that in computing the average score of Function args, all queries that do not involve a function call were filtered out.

Table 5 summarizes the mean score values for the function and argument selection, generated output, and overall evaluation, which is the average of the two individual metrics. It is clear that function selection, argument assignment, and response generation accuracy is intrinsically related to the complexity of the model. Larger models, such as Llama 3.3 (70B) and Mistral Small 3.1 (24B), achieve higher correctness due to their increased parameter count, greater depth, and enhanced attention mechanisms, which enable more expressive feature representations, improved contextual reasoning, and higher capacity for capturing complex dependencies. These attributes enhance their ability to capture contextual nuances, leading to more precise function retrieval and parameter selection. Additionally, their superior representational capacity improves response generation, enabling a more accurate interpretation of complex queries. Consequently, the observed trend confirms that increased model complexity translates into improved performance in function-related tasks.

However, there are instances where less complex models achieve performance on par with or even better than that of larger models. For example, in the function selection task, Llama 3.2 (3B) outperforms Llama 3.1 (8B), indicating that increased model size does not always correlate with a higher accuracy. This suggests that, for certain tasks, a smaller model can achieve comparable or superior performance while offering advantages in computational efficiency. This consideration is also supported by the function selection score exhibiting a significantly more gradual decline compared to the conversational output score as the model size decreases. Along with the previous observation, this trend suggests that function selection may be less dependent on extensive model parametrization, potentially allowing for the use of smaller models without a substantial loss in accuracy. In scenarios like LMD warehouse management, where real-time processing is critical, selecting a less complex model with competitive accuracy can provide a better balance between precision and speed. A hybrid approach could also be considered, where a smaller model is employed for function selection while a larger model handles conversational responses. However, the impact of such an approach on overall performance remains nontrivial due to the aforementioned caching mechanisms. As a result, while model specialization may theoretically optimize resource allocation, its practical benefits must be evaluated on a case-by-case basis in light of caching-induced efficiency gains.

### 4.3. Discussion

After a functional and performance evaluation focused on the technical aspects, it is important to consider how the overall proposal fits into the Research Gaps identified in Section 1. The analysis starts with key takeaways on the operational and economic benefits of the main technological elements of the proposed approach, reported in Table 6 along with potential limits and challenges for application to real LMD scenarios.

**RG1.** This work proposes a cohesive LMD framework combining data-driven AI-based optimization techniques for item allocation and fleet route management with user-centered interfaces exploiting Mixed Reality hands-free devices and LLM-based conversational assistants. In complex distributed systems, one of the main challenges is the integration of all components, which must include data sources such as sensor networks and legacy facilities. This proposal addresses this issue by adopting a microservice architecture with well-defined software interfaces and a Knowledge Graph component for data management. As elaborated on in Section 4.1, the latter grants uniformity and interoperability in information representation, context-aware retrieval, and support for reasoning capabilities more advanced than conventional approaches based on relational or NoSQL databases. In the same way, the adopted approach promotes interoperability with external systems, facilitating the integration of local LMD deployments with larger regional supply chain and logistics infrastructures based on machine-understandable information flows.

**RG2.** The proposed AI-based optimization methods for fleet route planning and warehouse resource allocation follow real-time modifications in problem input data, enhancing real-time decision-making in dynamic scenarios. This can grant operational benefits in terms of time and human error reduction, and also by virtue of integration with context-aware spatial and voice user interfaces. As discussed in Section 3.2, gathering useful data is a challenge in real implementations. In fact, data should be considered to come not only from sensing devices inside warehouses and in delivery vehicles, but also from, e.g., traffic data sensor networks for training predictive models integrated in the Route Planner.

**RG3.** The framework includes a RAG pipeline integrating general LLMs with a domain-specific KG. The approach enables contextual semantic reasoning in information retrieval from the KG. Additionally, it supports function selection and execution from user prompts to update the KG when the state of resources changes. Other available LMD frameworks incorporating LLMs lack this dynamic bidirectional information exchange capability. Nevertheless, careful LLM selection is a significant implementation challenge to achieve a satisfactory trade-off between efficiency and accuracy, as demonstrated by experiments in Section 4.2.

## 5. Conclusions and Future Work

This work presents a novel framework for Last-Mile Delivery in smart city contexts, integrating Artificial Intelligence-based fleet load optimization, Mixed Reality-assisted warehouse management, and a Knowledge Graph-driven Large Language Model chatbot for real-time decision support. The proposed approach enables seamless real-time information sharing, providing end-to-end visibility of the delivery process between fleet managers, who handle vehicle scheduling and route planning, and warehouse operators, who perform parcel pickup and vehicle loading. The framework enhances operational efficiency by integrating an OWL-based Knowledge Graph to structure and semantically enrich domain knowledge, ensuring consistency and interoperability across components. Additionally, a conversational assistant facilitates intuitive interaction with warehouse and fleet management roles, leveraging Retrieval Augmented Generation to provide context-aware responses. The scheduling and planning components extend open-source tools with customized problem configuration implementations, supporting continuous optimization meta-heuristics while incorporating an extensible AI-based traffic prediction module. Experimental results demonstrate the effectiveness of the LLM-based RAG pipeline, particularly highlighting the role of model complexity in function selection and response generation.

Future work will involve comprehensive performance comparisons of the proposed AI methods against state-of-the-art approaches using real and synthetic datasets for LMD scenarios. Further evaluations will be necessary to assess the system’s acceptability and increase its Technology Readiness Level (TRL) by identifying Key Performance Indicator (KPI) requirements and measuring outcomes in real-world field tests with logistics service providers. Additional research directions include extending the range of supported optimization and planning algorithms, as well as refining the integration of the KG and LLM to enhance contextual reasoning and query resolution. The incorporation of delivery drones in LMD operations is also under investigation, expanding fleet load optimization to mixed vehicle-drone routing problems [63], as well as improving drone path planning with onboard AI-driven flight corrections, and defining MR-based assistance procedures for drone preparation. Lastly, the framework will be extended with a third customization of the MR client module for couriers and with blockchain-based traceability functionalities, further enhancing transparency and security in Last-Mile Logistics.

## Figures and Tables

**Figure 1 sensors-25-02696-f001:**
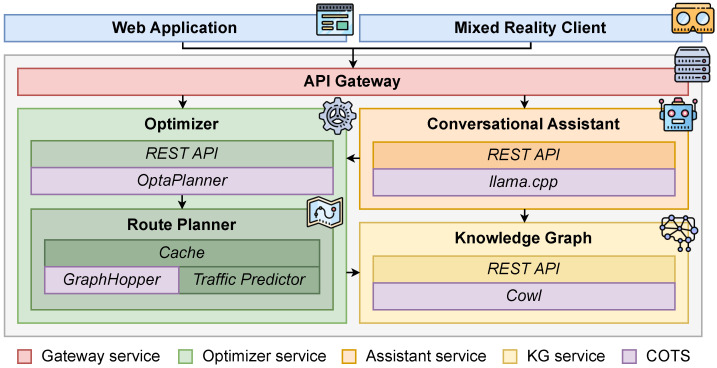
Logical components of the framework architecture.

**Figure 2 sensors-25-02696-f002:**
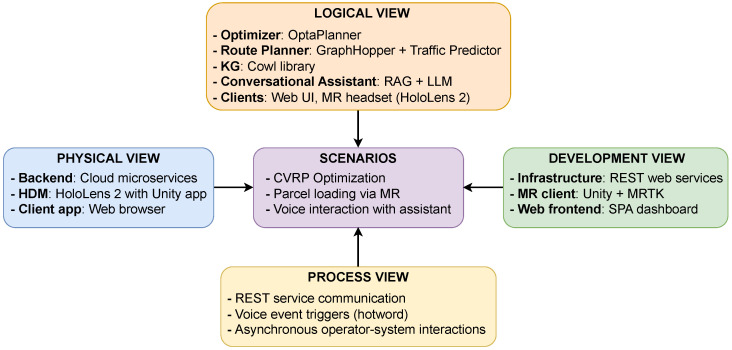
The 4 + 1 architectural view of the proposed framework.

**Figure 3 sensors-25-02696-f003:**
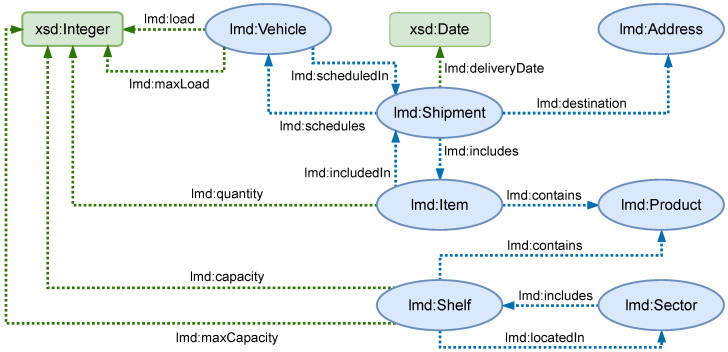
Reference graph-based data model.

**Figure 4 sensors-25-02696-f004:**
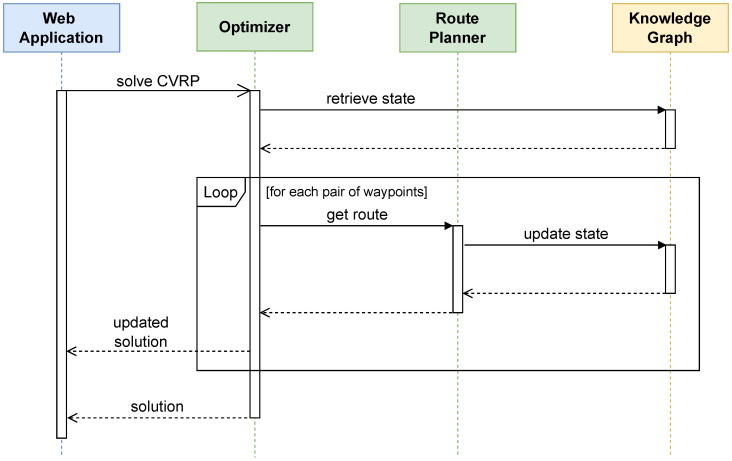
Interaction between the manager and the Optimizer service.

**Figure 5 sensors-25-02696-f005:**
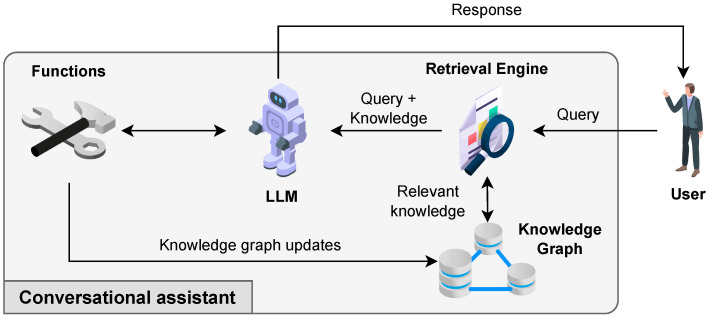
Architecture and data flow of the RAG pipeline.

**Figure 6 sensors-25-02696-f006:**
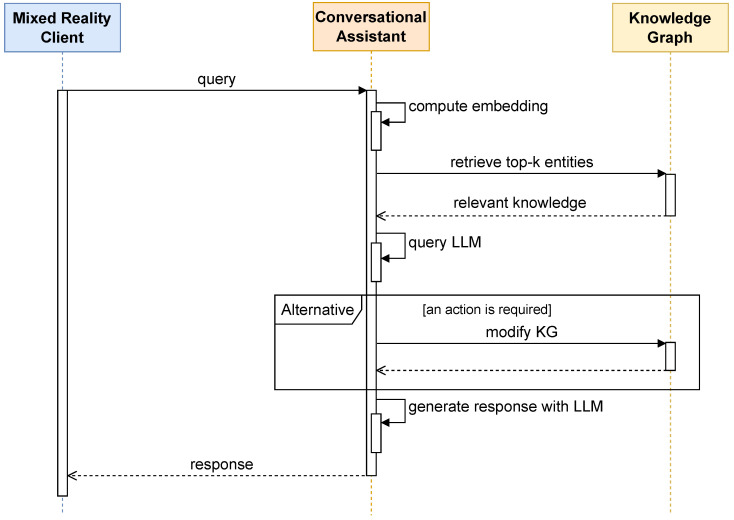
Interaction between the warehouse operator and the conversational assistant.

**Figure 7 sensors-25-02696-f007:**
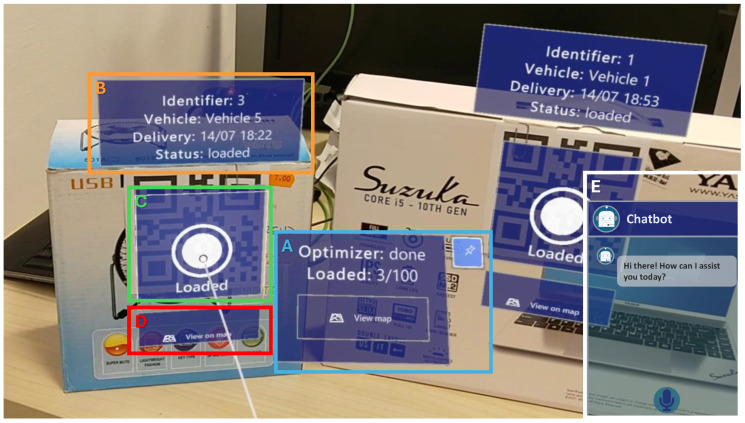
HoloLens 2 mixed reality client.

**Figure 8 sensors-25-02696-f008:**
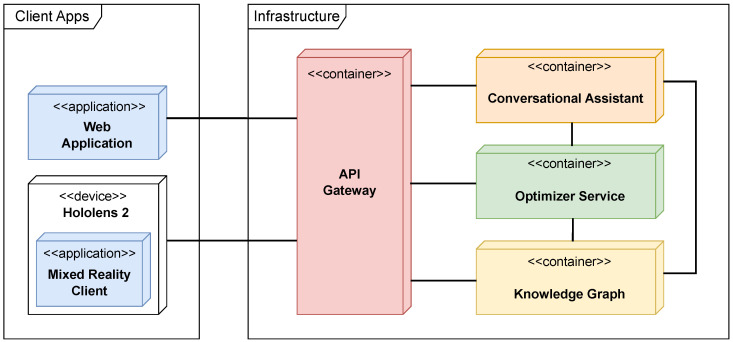
Deployment diagram.

**Figure 9 sensors-25-02696-f009:**
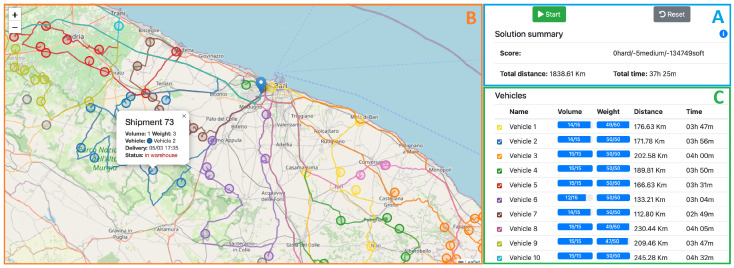
Last-Mile Delivery web application. A: Optimization process Start/Stop/Reset. B: Scheduled routes on the map. C: Optimized schedule.

**Figure 10 sensors-25-02696-f010:**
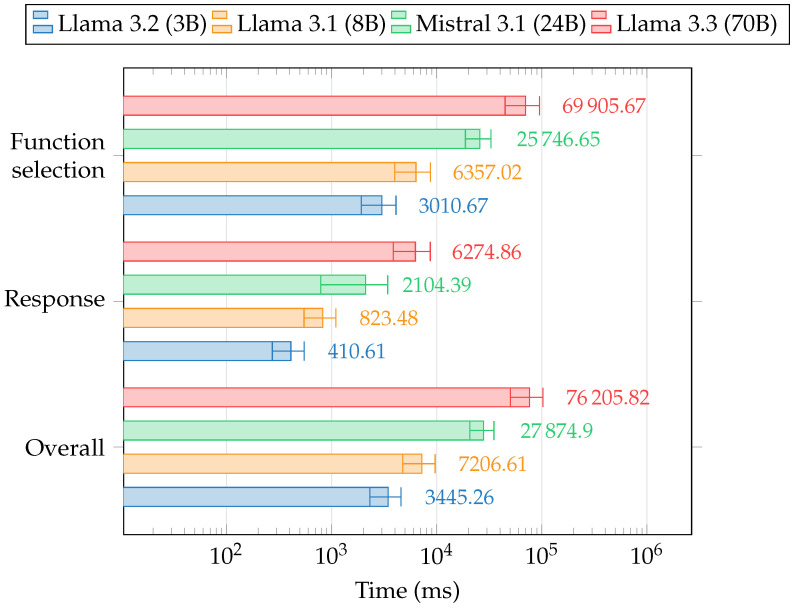
Averaged task-specific performance metrics for each LLM model.

**Figure 11 sensors-25-02696-f011:**
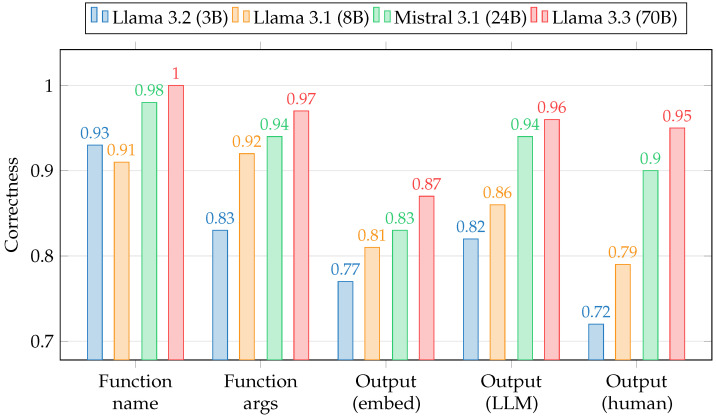
Mean correctness scores for each LLM model.

**Table 1 sensors-25-02696-t001:** Comparison of current approaches in Last-Mile Delivery. Capabilities include context-aware route planning (CRP), resource allocation optimization (RAO), and real-time operator support (ROS). ✓ Supported, ✗ Not Supported.

Ref.	CRP	RAO	ROS	AI Techniques	AR/VR Integration	Benefits	Challenges
**RG1. Lack of interoperable LMD frameworks including both AI algorithms and user-centered interfaces**							
[30]	✓	✗	✗	Exact and (meta-)heuristic methods	N/A	Well-established, computationally efficient.	Limited adaptability to dynamic scenarios.
[31]	✓	✗	✗	Deep reinforcement learning	N/A	Overall performance greater than most of the state-of-the-art heuristic methods.	Not suitable in the presence of time constraints, dynamic customer requests, or stochastic traffic conditions.
[33]	✓	✗	✗	Deep reinforcement learning	N/A	Great scalability, model efficient in solving instances of very large sizes.	Inaccurate results for small instances.
[32]	✓	✓	✗	ANNs for predicting delivery volumes	N/A	Reducing urban traffic by regulating the flow of vehicles.	Impact of external factors not considered when predicting the delivery time.
**RG2. Limited support for context-aware route planning and real-time decision-making in dynamic environments**							
[36]	✗	✓	✓	N/A	3D VR simulator for port logistics training	Improve efficiency and effectiveness of operations at ports and other logistics hubs.	No interactions between simulations and real processes.
[37]	✗	✗	✓	Image recognition	HMD for visualizing AR content	Support training sessions about picking activities for both novel and experienced operators.	Single-user use-cases and no interactions between AR applications and VRP models.
[38]	✗	✗	✓	N/A	AR visualization on Android devices	Assisted indoor navigation and path planning in medium-sized warehouses based on inertial sensors.	No additional information provided to the users regarding contextual conditions.
**RG3. Simplistic usage of LLMs, lacking integration of KG and edge computing technologies**							
[17]	✗	✗	✓	Retrieval-augmented generation (RAG)	N/A	Enhanced decision support through KG integration.	Integration complexity and security concerns, MR tools not supported.
[40]	✓	✗	✗	ChatGPT-3.5-based route optimization	N/A	Reduction in delivery distance.	Requires external validation due to inaccuracies in LLM responses.
[41]	✗	✗	✓	ChatGPT-3.5-based agentic framework	N/A	Machine-specific information retrieved through simple conversational assistants.	Results provided only in textual form, without any specific data modeling.
**This work**	✓	✓	✓	Exact and (meta-)heuristic optimization methods, ML-based traffic models, LLM-based assistant.	HMD for visualizing AR content and manipulating MR items	Integrated platform for fleet management and warehouse operations with real-time bidirectional information exchange, customizability of algorithms and models.	Prototype stage, performance evaluation required in real-world settings.

**Table 2 sensors-25-02696-t002:** Entities defined in the reference data model.

Entity	Description
Vehicle	A cargo vehicle in the delivery fleet; current and maximum loads are tracked.
Shipment	Delivery unit, which may contain one or more items and is assigned to a vehicle. It is associated to a destination address and a delivery date.
Address	Delivery destination.
Item	Line in a shipment, characterized by a product and a quantity.
Product	An instances of this class is an individual product stored in the distribution center and delivered to a final address.
Shelf	Warehouse’s smallest storage unit; can contain multiple products; current and maximum capacity are tracked.
Sector	Warehouse area; can include various shelves.

**Table 3 sensors-25-02696-t003:** Range of values for the random initialization procedure.

Item Volume	Item Weight	Vehicle Volume	Vehicle Weight
$1\leq x_{1}\leq 2$	$1\leq x_{2}\leq 10$	$x_{3}\leq 15$	$x_{4}\leq 50$

**Table 4 sensors-25-02696-t004:** LLM models used in the evaluation along with their key features.

Model Name	Parameters	Context Length	Embedding Size	Quantization
Meta Llama 3.2	3B	131,072	3072	Q4_K_M
Meta Llama 3.1	8B	131,072	4096	Q4_K_M
Mistral Small 3.1	24B	131,072	5120	Q4_K_M
Meta Llama 3.3	70B	131,072	8192	Q4_K_M

**Table 5 sensors-25-02696-t005:** Average correctness scores for function selection, output generation, and overall performance for each LLM model. Best scores are in bold, differences in color italics.

Model Name	Parameters	Function	Output	Overall
Llama 3.3	70B	**0.99**	**0.93**	**0.95**
Mistral 3.1	24B	0.96 *(−3%)*	0.89 *(−4%)*	0.92 *(−3%)*
Llama 3.1	8B	0.92 *(−7%)*	0.82 *(−12%)*	0.86 *(−9%)*
Llama 3.2	3B	0.88 *(−11%)*	0.77 *(−17%)*	0.81 *(−15%)*

**Table 6 sensors-25-02696-t006:** Technological solutions in the proposed LMD framework.

Technology	Operational Benefits	Economic Benefits	Challenges and Limitations
**AI forFleetManagement**	Data-driven route optimization Context-aware vehicle planning Reduced delivery delays	Reduced vehicle costs (e.g., fuel consumption) Reduced missed deliveries and penalties	Integration complexity for advanced scenarios Requires accurate and widespread sensor data collection
**AI forResourceAllocation**	Optimized item loading Reduced pick-up and drop-off times	Increased delivery throughput Maximized resource loading on vehicles	Requires real-time inventory and delivery data Integration with legacy systems
**MixedReality**	Hands-free task execution Multi-cue suggestions for complex tasks Faster operator training	Higher workforce productivity Reduced supervision costs	Requires devices for MR (e.g., HMDs)Requires user-centric application design
**KnowledgeGraph**	Context-aware data retrieval Reuse of well-known vocabularies Enables semantic reasoning	Reduced system integration costs due to improved interoperability	Data modeling based on Semantic Web guidelines Graph databases required for data storage
**LargeLanguageModels**	Natural language interactions Simplified information retrieval Improved decision support	Reduced onboarding and helpdesk costs	Prompt engineering techniques needed to avoid inaccurate responses Local deployment suggested for managing privacy-sensitive data

## Data Availability

Data available on request due to restrictions (e.g., privacy, legal or ethical reasons).

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
