# Peer review of "Enhancing Last-Mile Logistics: AI-Driven Fleet Optimization, Mixed Reality, and Large Language Model Assistants for Warehouse Operations"

_sensors, 2025, doi:10.3390/s25092696_

Round 1

Reviewer 1 Report

Comments and Suggestions for Authors

The manuscript addresses some challenges in LMD and highlights the use of AI to optimize fleet management and AR/MR to improve warehouse operations. The authors propose an architecture that combines AI-based courier assignment and vehicle routing with a KG-driven decision system and an LLM-based conversational assistant for managers and operators. The framework also includes an MR-based headset interface for warehouse operators to provide real-time data and interaction through virtual UI elements. Lastly, a case study/experiment (?) has been conducted. My comments on the manuscript are as follows:

  1. The number of articles cited in the introduction is very low. The authors should identify the research gap and tailor the proposed solution to fill that gap.
  2. If the goal of the research is to maximize the integration of technologies used in the warehouse with AI, why are other widely used assistive technologies in order picking, such as cobots and robots as the main component executing schedules generated by AI, not considered in the proposed architecture?
  3. The sentence "the system is implemented as a client-server platform leveraging Web technologies" brings to mind a two-tier client-server architecture, while the proposed architecture seems to be a multi-tier web-based design. I suggest the authors mention that they mean a client-server model of REST architecture (perhaps only in the application server), but the proposed architecture uses intermediate servers (like the API gateway, load balancer, web and database layers, etc.) to improve scalability and manageability without the client needing to know about them. However, a deployment diagram could better show this aspect of the system architecture.
  4. I suggest that the findings from the literature review be presented under several groups (e.g., order picking, routing, resource optimization, etc.) in Section 2. Authors could report these findings based on key concerns and decisions in LMD. See the following article on key concerns and decisions in LMD:

Alaeddini, M., & Asgari, A. (2025). An Integrated Big Data Analytics Architecture for Resilience: A Case Study of Last-Mile Agri-Food Delivery. Procedia Computer Science253, 1621-1630.

  1. Figure 1 should appear in Section 3, and Table 1 should appear in Section 2.
  2. For Section 3, I suggest modeling some views from the 4+1 view model as a good way to explain such a complex architecture. For example, a class diagram could be used to show the logical view (the structure of the system) by showing its classes, important attributes, operations, and the relationships between objects. For the development view, a component diagram could illustrate the organization and dependencies between software components, showing how the system is divided into microservices. Similarly, a deployment diagram could represent the physical view of the deployment of the system, showing how microservices are distributed across the infrastructure. In this way, the functionality for handling APIs is shown in the logical view, while the use of the REST protocol is shown in other views.
  3. Regarding Section 4, if the authors have performed only an experiment by simulation, they should replace the term "case study" with the term "experiment". It is also important to provide some information about the simulation environment and platform (hardware, software, virtualization, etc.).

Author Response

Reviewer 1

Q1.1 The number of articles cited in the introduction is very low. The authors should identify the research gap and tailor the proposed solution to fill that gap.
Response
We thank the reviewer for the feedback. We have revised the Introduction and Related Work sections to include a broader range of relevant works. We have also explicitly defined the research gaps emerging from the literature review and aligned them with the proposed solution to highlight how our work directly addresses these issues (see also the new discussion Section 4.3).

Q1.2 If the goal of the research is to maximize the integration of technologies used in the warehouse with AI, why are other widely used assistive technologies in order picking, such as cobots and robots as the main component executing schedules generated by AI, not considered in the proposed architecture?
Response
Thank you for this insightful remark. As clarified in the revised version of the manuscript, the proposed framework focuses on enhancing the interaction between human operators and AI-powered services through user-centered interfaces (leveraging mixed reality hands-free devices and voice-based conversational assistants). However, this does not exclude the integration of further assistive technologies, such as cobots and robots.
Section 3 explicitly highlights that the architecture is designed to be modular and extensible, supporting the addition of AI-based components such as robotic agents. By providing a machine-understandable representation of the state of the warehouse, the use of an OWL-based Knowledge Graph facilitates the exposure of services for Machine-to-Machine (M2M) interactions with automated agents.
This scalability is further discussed in Section 4.3, where we clarify that the proposed logic infrastructure allows seamless integration of autonomous devices and supports future scenarios involving robotic picking and/or actuation tasks.

Q1.3 The sentence ”the system is implemented as a client-server platform leveraging Web technologies” brings to mind a two-tier client-server architecture, while the proposed architecture seems to be a multi-tier web-based design. I suggest the authors mention that they mean a client-server model of REST architecture (perhaps only in the application server), but the proposed architecture uses intermediate servers (like the API gateway, load balancer, web and database layers, etc.) to improve scalability and manageability without the client needing to know about them. However, a deployment diagram could better show this aspect of the system architecture.
Response
We acknowledge the ambiguity of the sentence and we are grateful for this helpful observation. In the introduction of the revised version, we clarified that the proposed solution does not comply with a simplistic client-server architecture. As elaborated in the revised Section 3 and shown in Figure 1, it adopts a modular design based on several microservices accessed through a public Gateway RESTful API. This multi-layered approach allows the different logical components to be decoupled and deployed as single containerized services, enhancing both maintainability and scalability. This flexibility is also highlighted in the deployment diagram in the recently added Figure 8. Depending on workload requirements, the core services (Optimizer, KG, Conversational Assistant) can be executed on a single Virtual Machine instance, or they can be scaled and orchestrated using dedicated platforms (such as Kubernetes or Amazon ECS).

Q1.4 I suggest that the findings from the literature review be presented under several groups (e.g., order picking, routing, resource optimization, etc.) in Section 2. The authors could report these findings based on key concerns and decisions in LMD. See the following article on key concerns and decisions in LMD: Alaeddini, M., & Asgari, A. (2025). An Integrated Big Data Analytics Architecture for Resilience: A Case Study of Last-Mile Agri-Food Delivery. Procedia Computer Science, 253, 1621-1630.
Response
We have updated the Related Work section and Table 1 to accomplish the reviewer’s advice, cited works are now grouped according to the specific research gaps identified in the Introduction, as suggested in Q1.1. In addition, more recent studies have been included to provide a comprehensive view of the state of the art. We believe that this rearrangement clarifies how each study relates to our paper and to the gaps in the current literature, also highlighting its benefits and possible limitations.

Q1.5 Figure 1 should appear in Section 3, and Table 1 should appear in Section 2.
Response
Thank you for this remark, we have revised the manuscript accordingly. Figure 1 now appears in Section 3, and Table 1 has been moved to Section 2 to better align with the text.

Q1.6 For Section 3, I suggest modeling some views from the 4+1 view model as a good way to explain such a complex architecture. For example, a class diagram could be used to show the logical view (the structure of the system) by showing its classes, important attributes, operations, and the relationships between objects. For the development view, a component diagram could illustrate the organization and dependencies between software components, showing how the system is divided into microservices. Similarly, a deployment diagram could represent the physical view of the deployment of the system, showing how microservices are distributed across the infrastructure. In this way, the functionality for handling APIs is shown in the logical view, while the use of the REST protocol is shown in other views.
Response
We thank the reviewer for this valuable suggestion. In the revised version of the manuscript, Section 3 has been significantly updated according to the 4+1 architectural view model, in order to improve the clarity and completeness of the system description. In particular now:
•    Figure 2 summarizes the 4+1 architectural view, including the logical, development, process, and physical perspectives, along with representative usage scenarios.
•    The logical view is presented in Figure 1, which identifies the main software components and outlines the roles of Web and Mixed Reality clients. We have opted not to include a class diagram, as the structural aspects of the system are already covered by the ontology-based data model in Figure 3 and the logical decomposition in Figure 1.
•    The process view is illustrated through sequence diagrams that describe dynamic interactions. Specifically, Figure 4 outlines the interaction between the manager and the Optimizer service, while Figure 6 reports on the interaction between warehouse operators and the Conversational Assistant.
•    The development view, which highlights the modular organization of the system and the relationships among components, is presented in Figure 1 and is further discussed in Sections 3.4 and 4.
•    The physical view is represented in the deployment diagram in Figure 8, which shows how services are containerized and deployed across cloud or on-premise nodes, including edge devices such as the HoloLens HMD.

Q1.7 Regarding Section 4, if the authors have performed only an experiment by simulation, they should replace the term ”case study” with the term ”experiment”. It is also important to provide some information about the simulation environment and platform (hardware, software, virtualization, etc.).
Response
Yes, thank you. We agree. In the revised manuscript, we have replaced the term “case study” with ”experiment” in the Abstract and with “evaluation” in the title of Section 4, to better reflect the nature of the evaluation we carried out (which is based on a simulated scenario). To address the second part of the comment, we have added details about the simulation environment in Section 4.1, including information on the hardware and software platform used to deploy and test the system. Moreover, the Physical View diagram in Figure 8 reports on the deployment layout adopted during the simulation. This includes cloud/on-premise infrastructure and edge devices, clarifying how the system has been evaluated in a realistic, though simulated, operational environment.

Reviewer 2 Report

Comments and Suggestions for Authors

Introductory remarks:

The article presents the results on improving last mile logistics: AI-based fleet optimization, Mixed Reality and LLM assistants for warehouse operations on the example of a selected case study. The topic concerns areas important for researchers related to logistics, warehousing and transport, thanks to which it fits into the subject of the journal. The considerations refer to the current topics related to changes in the area of ​​implementation of solutions based on artificial intelligence in the logistics sector.

However, the research value of the article in its current form is significantly limited and in its current form for the year 2025 is not fully realistic. Both the research goals (there are no hypotheses) are not well defined, and their verification on the basis of the presented considerations is highly questionable. The structure of the article itself is not fully transparent. Similarly to the lack of correctness in the wording of the back of the article and numerous logical errors indicated OptaPlanner programs (considering their limitations - which the authors with such experience seem to know about) or others are not used in the field of last mile logistics - by the largest entities in this market sector, e.g. Amazon, or . The work is weakened by the lack of research input of the authors and an innovative approach to the topic based on the presentation of a practical solution based on artificial intelligence through the analysis of big data and neural networks - based on a large database of variable structure based on systems currently implemented by the largest operators in the world in the last mile logistics sector.

Although the work contains an example case study, this study is not fully representative - and the described tools cannot be an implementation tool in the last mile process in the practice of logistics activities - they can only be, for example, a solution when it comes to internal logistics of a given company that does not use outsourcing services or in a small area not related to large general cargo and reverse logistics. What the authors forgot to mention in their work. In the introduction, there is no reference to the subject of the essence of last mile logistics and the value chain, and to similar research results of other scientists based on other sources and related to this topic. Another significant shortcoming is the lack of a description of the evolution of changes in the definition of these concepts in the chapter devoted to the literature review - currently, in the era of so-called parcel lockers, drones, or the development of electromobility, or intelligent warehouses, this problem is more complicated. There is also no reference to the latest concepts described in the available literature. The research sample itself, research tools and the method of conducting research are very controversial, as is the method of inference - without proper description, justification and limitations. In addition, the work includes information on technological solutions and services related to e.g. telematics in logistics - in particular - which we are talking about, trying to identify potential benefits and barriers in relation to earlier systems. Therefore, it would be appropriate to use the SWOT method. Additionally, the work is weakened by the lack of reference to the economic aspects - key for the last mile. Citing statements in the text is not enough to draw general conclusions, such a procedure raises many doubts as to their methodological correctness. The presented discussion - polemics with other researchers - on the research results - is fragmentary in the work. A significant disadvantage is the fact that the presented conclusions are not confirmed by the presented research - concepts. The adopted research procedure and the scheme of the scientific process do not emphasize the innovativeness and contribution of the authors to science and their practical application - only statements were cited, omitting the analysis of cause-and-effect aspects. This leads to limited possibilities of scientific discussion with other researchers and low scientific value. The literature is too poor for such an extensive and often analyzed research topic. The reviewer states that the current form of the article requires thorough correction to meet the requirements set for works published in the MDPI journal concerning such a broadly defined research field in practice.

Detailed comments:

1) Consider changing the title. The title should refer to the content presented in the article. The content of the paper covers issues from several research areas in the field of last mile logistics, but the case study certainly does not fully apply artificial intelligence.

2) The abstract should be improved. Avoid detailed discussion of individual sections. The abstract should include the most important information on the context and background of the research - a possible research hypothesis, its purpose and procedures (choice of tools, description, assumptions, justification of measurements and methods, research dates, geographical area), key results and main conclusions, and contribution to the current state of knowledge. Practical application should be summarized.

3) Consider changing the keywords: last mile logistics, case study, process innovation, process optimization.

4) The introduction should clearly and concisely describe the background of the problem and the motives for undertaking the research. The purpose of the paper and possible research hypotheses should be verified and redefined. The considerations should be supplemented with a description of the essence of factors on innovations in last mile processes in logistics. The whole should be supplemented with references to the latest technological solutions concerning the last mile concept and the authors' contribution to filling the research gap. The authors' contribution to the literature should be emphasized, especially in the practical part. A breakdown of the article's structure should be presented with a description of the individual chapters.

5) The literature review should contain a synthetic presentation of research results from the latest published works on a given topic (or their chronological presentation of the discussed definitions, in tabular form) and a detailed assessment of the current state of knowledge on the subject and its gaps - it should also contain references to the latest studies on, for example, taking into account economic and technical aspects. For example, based on the SWOT method, which would constitute a real research contribution to the presented considerations. The whole should be transparent to every reader, even those who do not have specialist knowledge, such as the reviewer and authors.

6) Chapter 3 should be replaced by Material and methods and divided into individual sections: Conceptual assumptions, Sampling method, Tool description, Analysis scheme, which will significantly improve the clarity of the considerations:
It should be remembered that the presented content should be properly described and justified - the adopted criteria for selecting the research method, description, limitations, its innovativeness and the practical side of the method.

In particular, the description of the research should be conducted in such a way that it answers not only the questions posed in the article, but also allows for discussion with other researchers.

7) Distribution of results. It is necessary to refrain from quoting the authors' theses, especially since the article is to be of a research nature. Only the most important information should be highlighted and summarized to identify barriers and benefits - e.g. The reviewer proposes to present the model in the form of a scenario assessment - development barriers and benefits. or use another method of describing the obtained results. It would be reasonable to present the results, e.g. in tabular or graphical form. The whole should be supplemented with economic aspects. Such a research approach will allow for maintaining the methodological correctness of the research, formulating conclusions and starting a discussion.

8) Discussion of the chapter - it is necessary to develop the presented polemics by verifying the research results with other researchers, both in terms of a given geographical region and Europe. It is important to refer to the latest technological solutions and economic aspects from practice - in relation to the cited tools. The currently used inference method, based on clear and transparent criteria, is burdened with a methodological error. Because at present the conclusions do not result directly from the described research.

9) Application part. It should refer to the assumed goals of the work and the hypotheses for their verification. It should indicate new and important aspects of the conducted research, as well as a comprehensive interpretation of the obtained own results in relation to the current state of knowledge on a given topic - (this is currently missing, which weakens the research). Finally, the authors' contribution to the literature on the subject should be emphasized. The conclusions should be clear to every reader, even without specialist knowledge.

10) Please include an explanation of all abbreviations used in the article.

11) The list of literature should be supplemented with the latest literature on such a broadly presented research topic, with that of international scope.

12) The English language should be checked for correctness and use of technical vocabulary.

In summary, the reviewer appreciates the willingness of the authors to take on a very interesting and current topic. However, the article in its current form requires scientific refinement. For the good of both the authors themselves and the MDPI publishing house.

Comments on the Quality of English Language

The English language should be checked for correctness and use of technical vocabulary.

Author Response

Reviewer 2

Q2.1 However, the research value of the article in its current form is significantly limited and in its current form for the year 2025 is not fully realistic. Both the research goals (there are no hypotheses) are not well defined, and their verification on the basis of the presented considerations is highly questionable. The structure of the article itself is not fully transparent.

Response

Thanks for the comment: it give us the opportunity to clarify several aspects of our work. In particular, we have striven to make the research value of the paper more evident. Specifically, in the Introduction we have highlighted most relevant research gaps and the corresponding contributions the paper attempts to provide. Moreover, in Section 4.3 our proposal has been verified against the same research gaps and various considerations are made about provided benefits and challenges on implementing the work in real scenarios.

Q2.2 Similarly to the lack of correctness in the wording of the back of the article and numerous logical errors indicated OptaPlanner programs (considering their limitations - which the authors with such experience seem to know about) or others are not used in the field of last mile logistics - by the largest entities in this market sector, e.g. Amazon, or .

Response

We have now expanded references in both the Introduction and Related Work sections. They include relevant reports on the adoption of AI-based techniques by large entities in this market sector, such as FedEx, DHL Group and Amazon. Furthermore, in Section 3.2 we have reported on relevant use cases of OptaPlanner and GraphHopper from literature on problem instances of realistic size. We are grateful to the reviewer for the point.

Q2.3 The work is weakened by the lack of research input of the authors and an innovative approach to the topic based on the presentation of a practical solution based on artificial intelligence through the analysis of big data and neural networks - based on a large database of variable structure based on systems currently implemented by the largest operators in the world in the last mile logistics sector.

Response

In order to address this comment we have provided additional details on the scalability of artificial intelligence algorithms and tools included in the proposed framework. Hereinafter most relevant elements are summarized. (i) in Section 3.1, we have discussed on the Cowl library which supports ontologies with millions of axioms with lower memory usage than competing tools. (ii) in Section 3.2 reference examples on OptaPlanner and GraphHopper with problems of significant scale have been also referenced. In addition,

(iii) we have provided further details on the Traffic Predictor module: we have trained a predictive model of traffic flows based on a Convolutional Neural Network, using a dataset of over 8 million Origin-Destination flow records reported by a mobile operator over a 1-year period through analysis of SIM cellular coverage in the Apulia region.

Q2.4 Although the work contains an example case study, this study is not fully representative - and the described tools cannot be an implementation tool in the last mile process in the practice of logistics activities - they can only be, for example, a solution when it comes to internal logistics of a given company that does not use outsourcing services or in a small area not related to large general cargo and reverse logistics. What the authors forgot to mention in their work.

Response

Thanks to the reviewer for the remark. Although the proposed prototype and experiments focus on a singleoperator LMD scenario, we explicitly address this limitation in the revised manuscript by highlighting the potential of our KG-based framework to support end-to-end integration across different logistics actors. In particular, a new paragraph in Section 4.1 discusses how middle-mile logistics partners can publish semanticbased shipment metadata to the last-mile operator’s KG API. This enriched and machine-understandable representation allows the system to update in real time and enables warehouse operators to proactively schedule tasks and optimize resources.

Q2.5 In the introduction, there is no reference to the subject of the essence of last mile logistics and the value chain, and to similar research results of other scientists based on other sources and related to this topic. Another significant shortcoming is the lack of a description of the evolution of changes in the definition of these concepts in the chapter devoted to the literature review - currently, in the era of so-called parcel lockers, drones, or the development of electromobility, or intelligent warehouses, this problem is more complicated. There is also no reference to the latest concepts described in the available literature.

Response

Yes, right. We completely agree with the reviewer. To comply with the comment, now we have enriched both the Introduction and Related Work sections with references to the literature on last mile logistics. Also significant works presenting and assessing novel elements like parcel lockers, drones, electric mobility, and intelligent warehouses with their impact on last-mile delivery have been now referenced. We have also updated the analysis of the state-of-the-art works compared with our approach, by enriching it with significant recent contributions classified by topic. Everything can be found both in Table 1 and in the text of Section 2.

Q2.6 The research sample itself, research tools and the method of conducting research are very controversial, as is the method of inference - without proper description, justification and limitations.

Response

Following the reviewer’s remark we have striven to clarify our research approach in the paper. In particular, in the Introduction the most relevant research gaps from the literature have been now surveyed, and we have related them to our contribution. In addition, the differences among related works and the paper proposal have been clarified in Section 2. Furthermore, we have added a 4+1 architectural model view of our framework in Section 3, relating the various contributions of the work with each other from multiple perspectives. In Sections 3.1, 3.2 and 3.3 we have added further details and diagrams on the components of the proposed framework, while Sections 4, 4.1 and 4.2 of the revised manuscript provide more details on the developed prototype and experiments. Finally, Section 4.3 provides an analysis of the proposal with respect to the research gaps outlined in the Introduction, where operational and economic benefits are discussed along with possible limitations and implementation challenges in real scenarios.

Q2.7 In addition, the work includes information on technological solutions and services related to e.g. telematics in logistics - in particular - which we are talking about, trying to identify potential benefits and barriers in relation to earlier systems. Therefore, it would be appropriate to use the SWOT method. Additionally, the work is weakened by the lack of reference to the economic aspects - key for the last mile. Response

We thank the reviewer for the comment. Since our background is in computer science rather than management sciences, and our work’s contribution is mainly technological, we deem it best to refrain from a direct SWOT analysis or delving in the economic aspects systematically. However, an analysis in this perspective has been added to the revised manuscript by means of: (i) the discussion of main aspects of state-of-the-art AI-based LMD frameworks in Section 2 and what sets our work apart; (ii) the discussion of key operational and economic benefits of our approach (along with its limitations and implementation challenges) in Section 4.3, after the presentation of experimental results.

Q2.8 Citing statements in the text is not enough to draw general conclusions, such a procedure raises many doubts as to their methodological correctness. The presented discussion - polemics with other researchers on the research results - is fragmentary in the work. A significant disadvantage is the fact that the presented conclusions are not confirmed by the presented research - concepts. The adopted research procedure and the scheme of the scientific process do not emphasize the innovativeness and contribution of the authors to science and their practical application - only statements were cited, omitting the analysis of cause-and-effect aspects. This leads to limited possibilities of scientific discussion with other researchers and low scientific value. The literature is too poor for such an extensive and often analyzed research topic. The reviewer states that the current form of the article requires thorough correction to meet the requirements set for works published in the MDPI journal concerning such a broadly defined research field in practice.

Response

We are really grateful to the reviewer for this comment allowing us to point out that our intent was not the one of polemicizing with other researchers. We deeply apologize if our original manuscript left that impression. In order to mitigate this feeling, we have hardly worked to rearrange both Introduction and Related Work sections to frame our work in a more appropriate way. Specifically, we have striven to (1) enrich and update the corpus of referenced works, (2) provide a more accurate picture of the challenging research field of last-mile logistics, (3) include both research and industry contributions, (4) improve the discussion of approaches that have been compared to our proposal, (5) highlight what –in our opinion– distinguishes our proposal from existing works, without a confrontational tone.

Q2.9 Consider changing the title. The title should refer to the content presented in the article. The content of the paper covers issues from several research areas in the field of last mile logistics, but the case study certainly does not fully apply artificial intelligence.

Response

To the filed of Artificial Intelligence refers our prevailing background and most relevant aspects of our proposal. Nevertheless, considering this important remark and other reviewers’ advices, we definitively become aware that the contributions related to Artificial Intelligence were not exposed clearly enough in our original manuscript. For this reason, we have provided much more details in Sections 3 and 4 of the revised manuscript to highlight the AI-based elements, components and algorithms in our overall framework as well as in the prototype implementation for the case study. We are confident that the title matches the content of the manuscript much better now.

Q2.10 The abstract should be improved. Avoid detailed discussion of individual sections. The abstract should include the most important information on the context and background of the research - a possible research hypothesis, its purpose and procedures (choice of tools, description, assumptions, justification of measurements and methods, research dates, geographical area), key results and main conclusions, and contribution to the current state of knowledge. Practical application should be summarized.

Response

We thank the reviewer for the feedback. We have revised the abstract to better highlight the content of the paper and the core elements of the research.

Q2.11 Consider changing the keywords: last mile logistics, case study, process innovation, process optimization.

Response

Yes, we agree. We have updated the list of keywords according to the reviewer’s suggestion.

Q2.12 The introduction should clearly and concisely describe the background of the problem and the motives for undertaking the research. The purpose of the paper and possible research hypotheses should be verified and redefined. The considerations should be supplemented with a description of the essence of factors on innovations in last mile processes in logistics. The whole should be supplemented with references to the latest technological solutions concerning the last mile concept and the authors’ contribution to filling the research gap. The authors’ contribution to the literature should be emphasized, especially in the practical part. A breakdown of the article’s structure should be presented with a description of the individual chapters.

Response

Thanks for this remark. In the revised Introduction we have emphasized the contribution of the proposed work to address the existing research gaps in literature. Additionally, we have detailed the motivation for undertaking our research by citing recent studies that discuss the limitations of current LMD processes. Finally, a revised breakdown of the article’s structure has been defined, providing a detailed description of the content of each section.

Q2.13 The literature review should contain a synthetic presentation of research results from the latest published works on a given topic (or their chronological presentation of the discussed definitions, in tabular form) and a detailed assessment of the current state of knowledge on the subject and its gaps - it should also contain references to the latest studies on, for example, taking into account economic and technical aspects. For example, based on the SWOT method, which would constitute a real research contribution to the presented considerations. The whole should be transparent to every reader, even those who do not have specialist knowledge, such as the reviewer and authors.

Response

The Related Work section and Table 1 have been revised to improve the organization of cited studies, categorizing them based on the related research gaps. Furthermore, newer studies have been now incorporated to ensure a thorough representation of the current state of research. This restructuring enhances the clarity of how each study aligns with our research and addresses the identified gaps in the literature, while also highlighting the advantages and limitations of our proposal. Thank you for the advice.

Q2.14 Chapter 3 should be replaced by Material and methods and divided into individual sections: Conceptual assumptions, Sampling method, Tool description, Analysis scheme, which will significantly improve the clarity of the considerations: It should be remembered that the presented content should be properly described and justified - the adopted criteria for selecting the research method, description, limitations, its innovativeness and the practical side of the method. In particular, the description of the research should be conducted in such a way that it answers not only the questions posed in the article, but also allows for discussion with other researchers.

Response

We thank the reviewer for the valuable suggestions regarding the structure the paper, which inspired us to reorganize the manuscript to enhance clarity and logical flow. Specifically, we have restructured the Case Study and Experiments sections in a new section titled Evaluation, which now includes three dedicated

subsections: Prototype Deployment in a Realistic Scenario, Experimental Assessment of the RAG Pipeline, and Discussion. In addition, an improved introductory paragraph motivates the subsequent contents. The final discussion, in particular, aims to allow a debate on extant limitations and open challenges with other researchers. Section 3 has also undergone a deep content revision, providing a much more detailed analysis of the research methods and algorithms, along with a description of the novelty of the architecture and individual components, deployment strategies and potential limitations for practical applications. We hope that this revised organization better supports the articulation and justification of our methodological choices and provides a clearer response to the research hypotheses, particularly concerning the practical feasibility and performance of the proposed architecture in real-world last-mile logistics contexts.

Q2.15 Distribution of results. It is necessary to refrain from quoting the authors’ theses, especially since the article is to be of a research nature. Only the most important information should be highlighted and summarized to identify barriers and benefits - e.g. The reviewer proposes to present the model in the form of a scenario assessment - development barriers and benefits. or use another method of describing the obtained results. It would be reasonable to present the results, e.g. in tabular or graphical form. The whole should be supplemented with economic aspects. Such a research approach will allow for maintaining the methodological correctness of the research, formulating conclusions and starting a discussion.

Response

We have revised the manuscript to include a more structured presentation of the technologies and their implications. We have reworked the architectural diagram in Figure 1 for greater clarity and accuracy of presentation. Figure 2 has also been added to provide a systematic view of all the components of the proposed framework from multiple perspectives. The newly added Figures 4, 6 and 8 highlight interactions among the various components. Table 6 has been added to further highlight the operational and economic benefits, as well as the potential challenges associated with each technology applied in the proposed framework. Thanks to the reviewer for the suggestions.

Q2.16 Discussion of the chapter - it is necessary to develop the presented polemics by verifying the research results with other researchers, both in terms of a given geographical region and Europe. It is important to refer to the latest technological solutions and economic aspects from practice - in relation to the cited tools. The currently used inference method, based on clear and transparent criteria, is burdened with a methodological error. Because at present the conclusions do not result directly from the described research. Response

Following this comment, the new Section 4.3 now provides a discussion of the benefits and limitations of the technical and architectural contributions of the proposal in the perspective of the research gaps identified in the Introduction. This also matches the revised Related Work section, where the state of the art has been not only expanded –including regional and continental works from academia and industry– but also organized and discussed more clearly.

Q2.17 Application part. It should refer to the assumed goals of the work and the hypotheses for their verification. It should indicate new and important aspects of the conducted research, as well as a comprehensive interpretation of the obtained own results in relation to the current state of knowledge on a given topic - (this is currently missing, which weakens the research). Finally, the authors’ contribution to the literature on the subject should be emphasized. The conclusions should be clear to every reader, even without specialist knowledge.

Response

We really appreciate the reviewer’s feedback and her/his careful consideration of our work. We hope that the efforts made to address the previous comments, particularly Q2.12, Q2.13, Q2.14 and Q2.15 will contribute to improve the reliability and scientific value of the paper. We have revised the relevant sections to better discuss the novelty, practical contribution and alignment with current research trends.

Q2.18 Please include an explanation of all abbreviations used in the article.

Response

Thank you very much for this suggestion: we have included a comprehensive list of all acronyms at the end of the paper.

Q2.19 The list of literature should be supplemented with the latest literature on such a broadly presented research topic, with that of international scope.

Response

Yes, done: in the revised manuscript the overall number of references has more than doubled. Particularly in Sections 1 and 2 the discussion of literature background has been expanded (both geographically and in scope), updated and deepened significantly. Thank you.

Q2.20 The English language should be checked for correctness and use of technical vocabulary.

Response

We have carefully revised the English language and technical vocabulary to definitively improve them. Thank you very much.

Reviewer 3 Report

Comments and Suggestions for Authors

The paper presents an integrated AI-based framework for last-mile delivery. It combines fleet optimization, mixed reality interfaces, and LLM-powered assistants to enhance warehouse and logistics operations. The authors provide a case study in the Apulia region to demonstrate the practical deployment of their system. The study tackles a relevant and timely problem in logistics, particularly given the increasing complexity of urban deliveries. The integration of AI-driven optimization with knowledge graphs and MR for warehouse operations represents a novel contribution. However, while the paper presents a well-structured discussion, there are several areas that require significant revisions to strengthen the contribution and empirical validation.

  • The paper proposes an AI-integrated framework, but the novelty needs clearer articulation. The introduction should explicitly state how this approach differs from or improves upon existing AI-based LMD solutions.
  • The paper would benefit from a more direct comparison with recent works in AI-driven logistics and fleet management. A structured comparison table could help clarify this point.
  • The optimization approach leverages AI-based traffic prediction and heuristic methods, but the paper lacks sufficient details on hyperparameter selection and performance tuning.
  • The case study in the Apulia region is a valuable demonstration, but its generalizability remains unclear. How would this framework perform in other geographic or urban settings with different constraints?
  • The LLM-based assistant is a key component, but the evaluation results suggest variability in function accuracy across different LLM models. The authors should discuss the impact of different LLM models.

Comments on the Quality of English Language

The English quality of the paper is generally good, with clear explanations and well-structured arguments.

Author Response

Reviewer 3

Q3.1 The paper proposes an AI-integrated framework, but the novelty needs clearer articulation. The introduction should explicitly state how this approach differs from or improves upon existing AI-based LMD solutions.

Response

We thank the reviewer for this important comment. We have deeply revised the Introduction to more clearly articulate current research gaps and to emphasize the novelty of the proposed framework.

Q3.2 The paper would benefit from a more direct comparison with recent works in AI-driven logistics and fleet management. A structured comparison table could help clarify this point.

Response

The comparison table in the Related Work section has been extended to summarize the main aspects of the cited works. The table now includes key dimensions such as support for fleet operations, decisionmaking capabilities, interface modalities and the integration of AI-based tools. Furthermore, we have included additional recent works from the literature to provide an up-to-date overview of the research area as requested by the reviewer. This comparison highlights the distinctiveness of the proposed framework, which brings together the different components within a modular architecture. We are really grateful to this remark as it allows us to improve the paper self consistence.

Q3.3 The optimization approach leverages AI-based traffic prediction and heuristic methods, but the paper lacks sufficient details on hyperparameter selection and performance tuning.

Response

We are grateful for this insightful comment. To fulfill the request, we have expanded the manuscript including additional details about the architecture of the AI-based traffic prediction model, as well as the dataset used for its training. We are hopeful that this clarification strengthens the clarity of the overall optimization approach.

Q3.4 The case study in the Apulia region is a valuable demonstration, but its generalizability remains unclear. How would this framework perform in other geographic or urban settings with different constraints? Response

Thank you very much for the thoughtful observation. We have addressed this point in Section 4.1 by clarifying the framework’s adaptability to different geographic or urban contexts. Specifically, applying the framework to other regions primarily involves updating the underlying map data, which can be done with relatively low effort. Operational differences, such as those in warehouse processes, can be modeled through updates to the underlying knowledge graph, a key feature of our approach, with minimal programming effort. Finally, while the prototypical traffic predictor is tailored to the Apulia region, it can be replaced by any model trained on region-specific data, thanks to a generic integration interface. Moreover, the predictor itself is optional, as GraphHopper’s travel time estimates are automatically used as a fallback when no predictive model is available, making the system usable even if region-specific data are lacking.

Q3.5 The LLM-based assistant is a key component, but the evaluation results suggest variability in function accuracy across different LLM models. The authors should discuss the impact of different LLM models. Response

We fully agree that extending the range of evaluated LLMs enhances the robustness of the analysis. To accomplish this relevant point, we have carried out new experiments to include an additional model, Mistral Small 3.1 (24B), which offers an intermediate configuration between LLaMA 3.1 (8B) and LLaMA 3.3 (70B) in terms of parameters size. The results obtained with Mistral Small are consistent with those of the other models: its performance lies between LLaMA 3.1 8B and LLaMA 3.3 70B considering both inference time and output accuracy. This further supports our observations on the trade-offs between model size, latency, and inference correctness.

Round 2

Reviewer 1 Report

Comments and Suggestions for Authors

The quality of the manuscript has improved significantly. In my opinion, there is no obstacle to its publication. I congratulate the authors on their work.

Reviewer 2 Report

Comments and Suggestions for Authors

After proofreading, the reviewer does not make any comments to the text.
I congratulate the authors on their idea and wish them creative continuation of research in this field.

Reviewer 3 Report

Comments and Suggestions for Authors

The authors have answered my earlier comments in a satisfactory manner. As a result, I am happy to see the paper published in the journal.